# Rényi Neural Processes

**Xuesong Wang** [1]   **He Zhao** [1]   **Edwin V. Bonilla** [1]

## Abstract

Neural Processes (NPs) are deep probabilistic models that represent stochastic processes by conditioning their prior distributions on a set of context points. Despite their advantages in uncertainty estimation for complex distributions, NPs enforce parameterization coupling between the conditional prior model and the posterior model. We show that this coupling amounts to prior misspecification and revisit the NP objective to address this issue. More specifically, we propose Rényi Neural Processes (RNP), a method that replaces the standard KL divergence with the Rényi divergence, dampening the effects of the misspecified prior during posterior updates. We validate our approach across multiple benchmarks including regression and image inpainting tasks, and show significant performance improvements of RNPs in real-world problems. Our extensive experiments show consistently better log-likelihoods over state-of-the-art NP models.

## 1. Introduction

Neural processes (NPs) (Garnelo et al., 2018b) strive to represent stochastic processes via deep neural networks with desirable properties in uncertainty estimation and flexible feature representation. The vanilla NP (Garnelo et al., 2018b) predicts the distribution for unlabelled data given any set of observational data as *context*. The main advantage of NPs is to learn a set-dependent prior distribution, where the KL divergence is minimized between a posterior distribution conditioned on a *target* set with new data and the prior distribution conditioned on the context set (Kim et al., 2019; Jha et al., 2022; Bruinsma et al., 2023).

However, as the parameters of the conditional prior are unknown, NP proposes a coupling scheme where the model that parameterizes the prior distribution is forced to share

[1]CSIRO's Data61, Australia. Correspondence to: Xuesong Wang <xuesong.wang@data61.csiro.au>.

*Proceedings of the 42nd International Conference on Machine Learning*, Vancouver, Canada. PMLR 267, 2025. Copyright 2025 by the author(s).

its parameters with an approximate posterior model. In this paper we show that this parameter sharing amounts to prior misspecification and can yield a biased estimate of the posterior variance and deteriorate predictive performance (Cannon et al., 2022; Knoblauch et al., 2019). Such misspecification can be worsened under noisy context sets (Jung et al., 2024; Liu et al., 2024). Other cases of prior misspecification encompass domain shifts (Xiao et al., 2021), out-of-distribution predictions (Malinin & Gales, 2018) and adversarial samples (Stutz et al., 2019).

To address the prior misspecification caused by parameterization coupling in vanilla NPs, several studies have been proposed to relax the constraint (Wang et al., 2023; Wang & Van Hoof, 2022; Wu et al., 2018; Wicker et al., 2021). For instance, the prior and the posterior models can share partial parameters instead of the entire network (Rybkin et al., 2021; Liu et al., 2022); hierarchical latent variable models are also used where both models share the same global latent variable and induce prior or posterior-specific distribution parameterization (Shen et al., 2023; Requeima et al., 2019; Kim et al., 2021; Lin et al., 2021).

Instead of modifying the specifics of the NP model or its underlying architectures, in this paper we offer a new insight into prior misspecification in NPs through the lens of robust divergences (Futami et al., 2018), which seek to learn an alternative posterior without changing the parameters of interests. Instead of minimizing the standard KL divergence between the prior and posterior distributions, robust divergences are theoretically guaranteed to produce better posterior estimates under prior misspecification (Verine et al., 2024; Regli & Silva, 2018). The Rényi divergence (Li & Turner, 2016; Van Erven & Harremos, 2014b), for instance, introduces an additional parameter $\alpha$ to control how the prior distribution can regularize the posterior updates. This parameter allows us to reduce the regularization effects of the misspecified prior (Knoblauch et al., 2019), thereby mitigating the biased estimates of the posterior variance, avoiding oversmoothed predictions (Alemi et al., 2018; Higgins et al., 2017), and achieving performance improvements.

In light of this, we propose Rényi Neural Processes (RNPs) that focus on improving neural processes with a more robust objective. RNP minimizes the Rényi divergence between the posterior distribution defined on the target set and the true

posterior distribution given the context and target sets. We prove that RNP connects the common variational inference and maximum likelihood estimation objectives for training vanilla NPs via the hyperparameter $\alpha$, through which RNP provides the flexibility to dampen the effect of the misspecified prior and empower the posterior model for better expressiveness.

**Our main contributions** are summarized as:

1. We identify that the parameter coupling between the conditional prior and the approximate posterior inherent to NPs amounts to prior misspecification.

2. We introduce a new objective RNP that addresses this misspecification and that unifies the variational inference (VI) and maximum likelihood estimation (MLE) objectives.

3. We show that our RNP method can be applied to several SOTA NP families without changing the models, and provides generalization performance improvements over competing approaches [1].

## 2. Preliminaries

**Neural Processes**: Neural processes are a family of deep probabilistic models that represent stochastic processes (Wang & Van Hoof, 2020; Lee et al., 2020). Let $f_\tau : \mathbb{X} \to \mathbb{Y}$ be a function sampled from a stochastic process $p(f)$ where each $f_\tau$ maps some input features $\mathbf{x}$ to an output $\mathbf{y}$ and $\mathbb{D}_{\text{train}}$ and $\mathbb{D}_{\text{test}}$ are meta-tasks induced by different $f_{\text{train}}$ and $f_{\text{test}}$ during meta-training and meta-testing. For a specific task $\mathbb{D}_\tau$, we split the data further into a context set $\mathbb{C} : (X_C, Y_C) := \{(\mathbf{x}_m, \mathbf{y}_m)_{m=1}^M\}$ and a target set $\mathbb{T} : (X_T, Y_T) := \{(\mathbf{x}_n, \mathbf{y}_n)_{n=1}^N\} = \mathbb{D}_\tau \backslash \mathbb{C}$. Our goal is to predict the target labels given the target inputs and the observable context set: $p(Y_T | X_T, X_C, Y_C)$.

NPs (Garnelo et al., 2018b) introduce a latent variable $\mathbf{z}$ to parameterize the conditional distribution $p(f|\mathbb{C})$ and define the model as $p(Y_T|X_T, X_C, Y_C) = \int p_\theta(Y_T|X_T, \mathbf{z}) p_\varphi(\mathbf{z}|X_C, Y_C) d\mathbf{z}$ where $\theta$ and $\varphi$ are network parameters of the likelihood and prior (also known as recognition) models, respectively. Due to the intractable likelihood, two types of objectives including the variational inference (**VI**) and maximum likelihood (**ML**) estimation have been proposed to optimize the parameters (Foong et al., 2020; Nguyen & Grover, 2022; Bruinsma et al., 2023; Guo et al., 2023):

$$- \mathcal{L}_{VI}(\theta, \phi, \varphi) = \mathbb{E}_{\mathbb{D}_{\text{train}}}[\mathbb{E}_{q_\phi(\mathbf{z})} \log p_\theta(Y_T|X_T, \mathbf{z}) - D_{\text{KL}}(q_\phi(\mathbf{z}) \| p_\varphi(\mathbf{z}|X_C, Y_C))] \quad (1)$$

$$-\mathcal{L}_{ML}(\theta, \varphi) = \mathbb{E}_{\mathbb{D}_{\text{train}}}\left[\mathbb{E}_{p_\varphi(\mathbf{z}|X_C, Y_C)} \log p_\theta(Y_T|X_T, \mathbf{z})\right] \quad (2)$$

[1] Our code is published at https://github.com/csiro-funml/renyineuralprocesses

The approximate posterior distribution for VI-based methods is usually chosen as $q_\phi(\mathbf{z}) = q_\phi(\mathbf{z}|\mathbb{C}, \mathbb{T})$. As the parameters of the conditional prior $p_\varphi(\mathbf{z}|X_C, Y_C)$ are unknown, NPs couple its parameters with the approximate posterior $p_\varphi(\mathbf{z}|X_C, Y_C) \approx q_\phi(\mathbf{z}|X_C, Y_C)$. We now replace the notation of the approximate posterior with $\varphi$ to be consistent with the ML objective:

$$- \mathcal{L}_{VI}(\theta, \varphi) \approx \mathbb{E}_{\mathbb{D}_{\text{train}}}[\mathbb{E}_{q_\varphi(\mathbf{z})} \log p_\theta(Y_T|X_T, \mathbf{z}) - D_{\text{KL}}(q_\varphi(\mathbf{z}|\mathbb{C}, \mathbb{T}) \| q_\varphi(\mathbf{z}|\mathbb{C}))] \quad (3)$$

The KL term in Eq 3 is sometimes referred to as the **consistency regularizer** (Wang et al., 2023; Foong et al., 2020), which assumes the parameter coupling between the conditional prior and the posterior. This assumption, as will show later, is the source of the inference suboptimality of vanilla NPs in the existence of finite context data.

**Rényi Divergences**: The Rényi divergence (**RD**) (Van Erven & Harremos, 2014a; Rényi, 1961) is defined on two distributions with a hyperparamter $\alpha \in (0, +\infty)$ and $\alpha \neq 1$:

$$D_\alpha(q(\mathbf{z}) \| p(\mathbf{z})) = \frac{1}{\alpha - 1} \log \int q(\mathbf{z})^\alpha p(\mathbf{z})^{1-\alpha} d\mathbf{z}$$
$$= \frac{1}{\alpha - 1} \log \mathbb{E}_{q(\mathbf{z})} \left[\frac{p(\mathbf{z})}{q(\mathbf{z})}\right]^{1-\alpha} \quad (4)$$

Note that the RD is closely related to the KL divergence in that if $\alpha \to 1$ then $D_\alpha(q\|p) \to D_{\text{KL}}(q\|p)$ (Van Erven & Harremos, 2014a). In other words, choosing $\alpha$ close to 1 for variational inference would result in a posterior as close to the standard VIs. Changing the KL divergence to RD can induce a robust posterior via the hyperparameter $\alpha$, as the influence of the density ratio $\frac{p}{q}$ determines how much we can penalize the posterior with the prior. With the flexibility of choosing $\alpha$, the model can adjust the degree of prior penalization. When the prior is misspecified, choosing the RD can lead to a more robust posterior that focuses more on improving the likelihood and less on prior regularization (Futami et al., 2018; Regli & Silva, 2018).

## 3. Rényi Neural Processes

In this section we describe our Rényi Neural Process (RNP) framework, a simple yet effective strategy which provides a more robust way to learn neural processes without changing the model. We start by analyzing the main limitation of the standard neural process objective and present a motivating example. We illustrate a case where the prior is misspecified and describe our new objective with the RD to mitigate this.

### 3.1. Motivation: Standard neural processes and prior misspecification

**Definition 3.1.** (Prior misspecification (Huang et al., 2024)) Let $q_\varphi(\mathbf{z}|X, Y)$ be a distribution associated with a map $\mathcal{X} \times \mathcal{Y} \to \mathcal{Z}$ parameterized by $\varphi$ and $p(\mathbf{z}|X, Y)$ is the ground truth distribution. Then, $\{q_\varphi(\mathbf{z}|X, Y), \varphi \in \Phi\}$ defines a

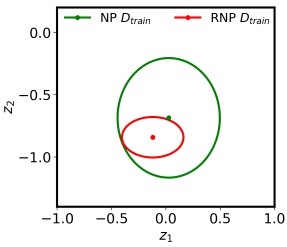

(a) Posterior distributions.

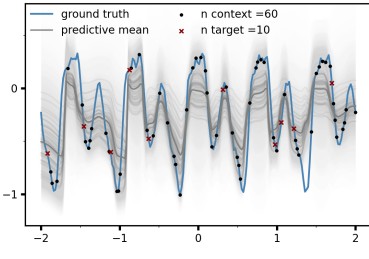

(b) Predictions of ANP using $\mathcal{L}_{VI}$

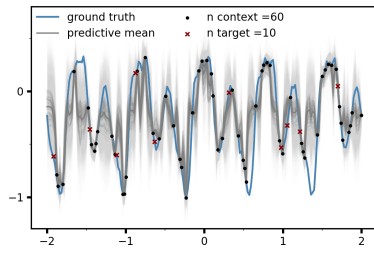

(c) Predictions using $\mathcal{L}_{RNP}$

*Figure 1.* An illustrative example. (a) Two 2D Gaussian posteriors $q(\mathbf{z}|\mathbb{C})$ conditioned on the context set using $\mathcal{L}_{NP}$ VI objective and our $\mathcal{L}_{RNP}$. (b) Predictive results on a GP regression dataset using the VI objective. (c) Results obtained with our RNP objective.

set of distributions on the space $\mathcal{Z}$ induced by the model. The prior model is misspecified if $\forall \varphi \in \Phi, q_\varphi(\mathbf{z}|X,Y) \neq p(\mathbf{z}|X,Y)$.

This translates to NPs as the approximate prior model $q_\varphi(\mathbf{z}|X_C, Y_C)$ in Eq 3 can not recover the ground truth prior $p(\mathbf{z}|X_C, Y_C)$ for any parameterization of $\varphi$. We now show how this definition can assist us to analyze how a misspecified prior model can hinder neural processes learning.

**Proposition 3.2.** *Due to the coupling parameterization of $\varphi$ in Eq 3, the NP prior model can be misspecified, which is unable to bound the ground truth marginal likelihood $p(Y_T|X_T, X_C, Y_C)$.*

Detailed proof can be found in Supp A.4. We state the main finding here: the prior term in the ELBO gradient estimate of posterior parameters is approximated in $\mathcal{L}_{VI}$: $\mathbb{E}_{q_\varphi(\mathbf{z}|\mathbb{C},\mathbb{T})} \nabla_\varphi \log p(\mathbf{z}|\mathbb{C}) \approx \mathbb{E}_{q_\varphi(\mathbf{z}|\mathbb{C},\mathbb{T})} \nabla_\varphi \log q_\varphi(\mathbf{z}|\mathbb{C})$. Due to this parameter coupling, we now have a "learned" prior model $q_\varphi(\mathbf{z}|\mathbb{C})$ that can move towards posterior samples $\mathbf{z} \sim q_\varphi(\mathbf{z}|\mathbb{C},\mathbb{T})$ which can lead to a biased estimate of the posterior and peculiar optimization dynamics. We will later show that by using our objective, this biased term is scaled by the data likelihood such that the gradient is smaller if the posterior is not likely to generate the data, therefore dampening the effects of prior misspecification.

**Illustrative example**: We first use Fig 1 to compare the posteriors and predictions of neural processes using two objectives. *The objective of this example is to compare how the vanilla NP and our to-be-proposed RNP behave when the prior is misspecified*. Both prior models are potentially misspecified due to the coupling parameterization scheme. We delay the introduction of the formulation of $\mathcal{L}_{RNP}$ in the next section, which is unnecessary for the illustration.

The resulted posteriors in Fig 1(a) show that RNPs obtain a much smaller variance estimate, suggesting that the $D_{\mathrm{KL}}(q_\varphi(\mathbf{z}|\mathbb{T},\mathbb{C})\|q_\varphi(\mathbf{z}|\mathbb{C}))$ is too strong in the vanilla NPs. As a result, vanilla NPs produce oversmoothed predictions (Fig 1(b) with a large variance that underfits the data). In contrast, our RNP dampens the impacts of prior misspecification and produces smaller variance but more expressive

predictions as shown in Fig 1(c). We will now introduce our new neural process learning method, which can be applied not only to the VI objective, but also the ML objective where SOTA NPs were trained with.

### 3.2. Proposed Method: Neural processes with the new objectives

**New objective for VI-based NPs.** The main issue of NPs is the prior approximation $q_\varphi(\mathbf{z}|X_C, Y_C)$ wrt the true prior $p(\mathbf{z}|X_C, Y_C)$. In this case, the posterior variance may be critically overestimated in some regions and underestimated in others. We therefore seek to obtain an alternative posterior distribution to alleviate this prior misspecification. Depending on whether the original NP framework is trained using the VI or the ML objective, we can revise the objective by minimizing the RD instead of KLD on two distributions. More specifically, in the case of VI-NPs where the inference of the latent variable $q(\mathbf{z})$ is of interest, the RD between the the approximated posterior distribution $q_\varphi(\mathbf{z}|X_T, Y_T, X_C, Y_C) = q_\varphi(\mathbf{z}|\mathbb{C},\mathbb{T})$ and the true posterior $p(\mathbf{z}|X_T, Y_T, X_C, Y_C) = p(\mathbf{z}|\mathbb{C},\mathbb{T})$ is minimized:

$$\min_{\theta,\varphi} D_\alpha \left( q_\varphi(\mathbf{z}|\mathbb{C},\mathbb{T}) \| p(\mathbf{z}|\mathbb{C},\mathbb{T}) \right)$$

$$\approx \max_{\theta,\varphi} \frac{1}{1-\alpha} \log \mathbb{E}_{q_\varphi(\mathbf{z}|\mathbb{C},\mathbb{T})} \left[ \frac{p_\theta(Y_T|X_T, \mathbf{z}) q_\varphi(\mathbf{z}|\mathbb{C})}{q_\varphi(\mathbf{z}|\mathbb{C},\mathbb{T})} \right]^{1-\alpha},$$
(5)

where the details can be found in A.5. Eq 5 is an approximation obtained by replacing $p(\mathbf{z}|X_C, Y_C)$ with $q_\varphi(\mathbf{z}|X_C, Y_C)$. We can approximate the intractable expectation with Monte Carlo:

$$-\mathcal{L}_{RNP}(\theta, \varphi) = \frac{1}{1-\alpha} \mathbb{E}_{\mathbb{D}_{\mathrm{train}}} \log \frac{1}{K} ($$

$$\sum_{k=1}^{K} \left[ \frac{p_\theta(Y_T|X_T, \mathbf{z}_k) q_\varphi(\mathbf{z}_k|\mathbb{C})}{q_\varphi(\mathbf{z}_k|\mathbb{C},\mathbb{T})} \right]^{1-\alpha} ),$$
(6)

where $\mathbf{z}_k \sim q_\varphi(\mathbf{z}|\mathbb{C},\mathbb{T})$

**Tuning $\alpha \in (0,1)$ can mitigate prior misspecification in RNPs**. We will present how RNP can handle prior misspecification through the gradients of the parameters of the encoder networks $\varphi$ (more details can be found in Supp A.8):

$$\nabla_\varphi \mathcal{L}_{RNP} = \sum_{k=1}^{K} \left( \frac{w_k^{1-\alpha}}{\sum_{k=1}^{K} w_k^{1-\alpha}} \nabla_\varphi \log w_k \right), \qquad (7)$$

$$\text{where } w_k = \frac{p_\theta(Y_T|X_T, \mathbf{z}_k) q_\varphi(\mathbf{z}_k|\mathbb{C})}{q_\varphi(\mathbf{z}_k|\mathbb{C}, \mathbb{T})}, \ \mathbf{z}_k \sim q_\varphi(\mathbf{z}|\mathbb{C}, \mathbb{T}) \quad (8)$$

Note that $\nabla_\varphi \log w_k$ corresponds to the gradient of the $\mathcal{L}_{VI}$ objective. As Daudel et al. (2023) pointed out, Rényi variational inference amounts to an importance weighted variational inference where the weight $w_k$ is scaled by the power of $(1-\alpha)$. Therefore, the prior gradient inside the log term $\nabla_\varphi \log q_\varphi(\mathbf{z}|\mathbb{C})$ is also scaled by the weight $\frac{w_k^{1-\alpha}}{\sum_{k=1}^{K} w_k^{1-\alpha}}$ where high-likelihood samples are given bigger weights whereas low-likelihood samples are given smaller weights. As a result, even if the prior is misspecified, as long as the the weight is small, it will not affect posterior update. We can control how much we trust the likelihood weight to guide the posterior by tuning the $\alpha \in (0, 1)$. When $\alpha = 1$ and $k = 1$ the RNP will recover the $\mathcal{L}_{VI}$ behavior. When $\alpha = 0$ the RNP recovers the maximum likelihood behavior as introduced next.

**New objective for ML-based NPs.** As the goal of NPs is to maximize the predictive likelihood instead of inferring the latent distribution, another type of NPs directly parameterizes the likelihood model without explicitly defining the latent variable $\mathbf{z}$. Following Futami et al. (2018), we can rewrite the maximum likelihood estimation as minimizing the KLD between the empirical distribution $\hat{p}(\mathbf{y}|\mathbf{x}, \mathbb{C})$ and the model distribution $p(\mathbf{y}|\mathbf{x}, \mathbb{C}, \theta)$:

$$-\mathcal{L}_{ML}(\theta) = \max_\theta \mathbb{E}_{\mathbb{D}_{\text{train}}} \log p_\theta(Y_T|X_T, \mathbb{C})$$

$$\equiv \max_\theta \mathbb{E}_{\mathbb{D}_{\text{train}}} \left[ \frac{1}{N} \sum_{n=1}^{N} \log p_\theta(\mathbf{y}_n|\mathbf{x}_n, \mathbb{C}) \right] \quad (9)$$

$$\approx \min_\theta \mathbb{E}_{\mathbb{D}_{\text{train}}} \left[ D_{\text{KL}}(\hat{p}(\mathbf{y}|\mathbf{x}, \mathbb{C}) \| p_\theta(\mathbf{y}|\mathbf{x}, \mathbb{C})) \right],$$

where $\hat{p}(\mathbf{y}|\mathbf{x}, \mathbb{C})$ is the empirical distribution defined as $\frac{1}{N} \sum_{n=1}^{N} \delta(\mathbf{y}, \mathbf{y}_n)$ where $y_n$ are samples from the unknown distribution $p^*(\mathbf{y}|\mathbf{x}, \mathbb{C})$. Replacing the KLD with RD gives:

$$\min_\theta \mathbb{E}_{\mathbb{D}_{\text{train}}} D_\alpha (\hat{p}(\mathbf{y}|\mathbf{x}, \mathbb{C}) \| p_\theta(\mathbf{y}|\mathbf{x}, \mathbb{C}))$$

$$\approx \min_\theta \mathbb{E}_{\mathbb{D}_{\text{train}}} \frac{1}{N} \sum_{n=1}^{N} \frac{1}{\alpha - 1} \log p_\theta^{1-\alpha}(\mathbf{y}_n|\mathbf{x}_n, \mathbb{C}) + Const$$

$$(10)$$

$$\mathcal{L}_{RNPML}(\theta) = \mathbb{E}_{\mathbb{D}_{\text{train}}} \frac{1}{(\alpha - 1)N} \sum_{n=1}^{N} \log p_\theta^{1-\alpha}(\mathbf{y}_n|\mathbf{x}_n, \mathbb{C})$$

$$= \mathbb{E}_{\mathbb{D}_{\text{train}}} \frac{1}{(\alpha - 1)N} \sum_{n=1}^{N} \log \left( \int p_\theta(\mathbf{y}_n, \mathbf{z}|\mathbf{x}_n, \mathbb{C}) d\mathbf{z} \right)^{1-\alpha},$$

$$(11)$$

where details can be found in A.6. Note that $\alpha = 0$ corresponds to the maximum likelihood estimation and

the new RNP objective essentially reweights the samples based on their likelihood. We define $p_\theta(\mathbf{y}_n, \mathbf{z}|\mathbf{x}_n, \mathbb{C}) = p_\theta(\mathbf{y}_n, \mathbf{z}|\mathbf{x}_n) p(\mathbf{z}|\mathbb{C})$ and use $p_\varphi(\mathbf{z}|X_C, Y_C) \approx p(\mathbf{z}|\mathbb{C})$. Note that this prior can still be misspecified in NN parameters of $\varphi$ and the family of distributions we choose for the approximation distribution $p$. Then Eq 11 can be approximated by Monte Carlo:

$$\mathcal{L}_{RNPML}(\theta, \varphi) \approx \mathbb{E}_{\mathbb{D}_{\text{train}}} \frac{1}{(\alpha - 1)N} \sum_{n=1}^{N} \log($$

$$\frac{1}{K} \sum_{k=1}^{K} p_\theta(\mathbf{y}_n|\mathbf{z}_k, \mathbf{x}_n))^{1-\alpha}, \text{ where } \mathbf{z}_k \sim p_\varphi(\mathbf{z}|\mathbb{C})$$

$$(12)$$

One advantage of ML-based method is that we do not need to estimate the density of the samples from the prior model. Hence, reparameterization tricks can be applied to obtain samples from non-standard distributions: $\mathbf{z}_k = s_\varphi(X_C, Y_C, \epsilon_k), \epsilon_k \sim \mathcal{N}(\mathbf{0}, I)$ with a neural network $s_\varphi$.

### 3.3. Properties of RNPs

We generalize the theorem from Rényi variational inference (Li & Turner, 2016) to RNPs:

**Theorem 3.3.** *(Monotonicity (Li & Turner, 2016))*

$\mathcal{L}_{RNP}$ *is continuous and non-increasing with respect to the hyper-parameter* $\alpha$.

**Proposition 3.4.** *(Unification of the objectives)*

$$\mathcal{L}_{ML} = \mathcal{L}_{RNP,\alpha=0} \geq \mathcal{L}_{RNP,\alpha\in(0,1)} \geq \mathcal{L}_{RNP,\alpha\to1} = \mathcal{L}_{VI}.$$

Theorem 3.3 and Proposition 3.4 (Proof see Supp A.7) bridge the two commonly adopted objectives via our RNP framework. We show that these objectives are bounded by $\log \int p(Y_T|X_T, \mathbf{z}) q(\mathbf{z}|X_C, Y_C)$. Although it is not the marginal due to the approximation $q(\mathbf{z}|X_C, Y_C) \approx p(\mathbf{z}|X_C, Y_C)$, the gradient of the RNP is more robust against misspecified priors and hence improve the posterior update.

### 3.4. Prior, posterior and likelihood models

One main advantage of RNP is we do not need to change the parameters of interests of the original NP models. Therefore, for the likelihood model $p_\theta(Y_T|X_T, \mathbf{z})$ we can adopt simple model architectures like NPs (Garnelo et al., 2018b) which assume independence between target points $p_\theta(Y_T|X_T, \mathbf{z}) = \prod_{n=1}^{N} p_\theta(\mathbf{y}_n|\mathbf{x}_n, \mathbf{z})$. The distribution of each target point is then modeled as Gaussian $p_\theta(\mathbf{y}_n|\mathbf{x}_n, \mathbf{z}) = \mathcal{N}(h_\mu(\mathbf{x}_n, \mathbf{z}), \text{Diag}(h_\sigma(\mathbf{x}_n, \mathbf{z})))$, and the decoder networks $h_\mu$ and $h_\sigma$ map the concatenation of the input feature $\mathbf{x}_n$ and $\mathbf{z}$ to the distribution parameters.

The prior model $q_\varphi(\mathbf{z}|X_C, Y_C)$ is more interesting as it is a set-conditional distribution and we are supposed to sample from it and evaluate the density of the samples.

One feasible solution is to define a parametric distribution on a DeepSet (Zaheer et al., 2017). For instance, $q_\varphi(\mathbf{z}|X_C, Y_C) = \mathcal{N}(g_\mu(h(\mathbb{C})), \text{Diag}(g_\sigma(h(\mathbb{C}))))$ where $h(\mathbb{C}) = \frac{1}{|\mathbb{C}|} \sum_{m=1}^{|\mathbb{C}|} h(\mathbf{x}_m, \mathbf{y}_m)$ is a DeepSet function on the context set $\mathbb{C}$. In practice diagonal Gaussian distributions worked well with high dimensional latent variables $\mathbf{z}$. ANPs (Kim et al., 2019) incorporate dependencies between context points $q_\varphi(\mathbf{z}|\mathbb{C}) = q_\varphi(\mathbf{z}|\mathbf{x}_{1:m}, \mathbf{y}_{1:m})$ using self-attention networks. But one can consider more flexible distributions such as conditional normalizing flows (Luo et al., 2023) for sample and density estimation. As previously stated, the posterior distribution is defined by coupling its parameters with the prior. Therefore the posterior $q_\varphi(\mathbf{z}|X_T, Y_T, X_C, Y_C)$ in the DeepSet case can be represented as $q_\varphi(\mathbf{z}|X_T, Y_T, X_C, Y_C) = \mathcal{N}(g_\mu(h(\mathbb{C}, \mathbb{T})), \text{Diag}(g_\sigma(h(\mathbb{C}, \mathbb{T}))))$. To apply stochastic gradient descent over the parameters of the posterior, we applied the reparameterization trick to obtain samples $\mathbf{z}_k = g_\sigma^{\frac{1}{2}}(h(\mathbb{C}, \mathbb{T})) * \epsilon + g_\mu(h(\mathbb{C}, \mathbb{T})), \epsilon \sim \mathcal{N}(\mathbf{0}, \mathbf{I})$.

### 3.5. Inference with Rényi Neural Processes

During inference time, as we cannot access the ground truth for the target outputs $Y_T$, we use the approximate prior $q(\mathbf{z}|X_C, Y_C)$ instead of the posterior distribution $q(\mathbf{z}|X_T, Y_T, X_C, Y_C)$ to estimate the marginal distribution:

$$p(Y_T|X_T, X_C, Y_C) = \int p_\theta(Y_T|X_T, \mathbf{z}) q_\varphi(\mathbf{z}|X_C, Y_C) d\mathbf{z}$$
$$\approx \frac{1}{K} \sum_{k=1}^K p_\theta(Y_T|X_T, \mathbf{z}_k), \mathbf{z}_k \sim q_\varphi(\mathbf{z}|X_C, Y_C). \tag{13}$$

We now provide the pseudo code for Rényi Neural Processes in Supp Algorithm 1. In addition to vanilla neural processes, our framework can also be generalized to other neural process variants as shown in the experiments section.

## 4. Related Work

**Neural processes family.** Neural processes (Garnelo et al., 2018b) and conditional neural processes (Garnelo et al., 2018a) were initially proposed for the meta learning scheme where they make predictions given a few observations as context. Both of them use deepset models (Zaheer et al., 2017) to map a finite number of data points to a high dimensional vector and their likelihood models assume independencies among data points. The main difference is whether estimating likelihood maximization directly or introducing the latent variable and adopting variational inference framework. Since their introduction, Neural Processes have seen widespread adoption in various fields, including continuous learning (Jha et al., 2024), spatial temporal forecasting (Wang et al., 2021) and weather prediction (Allen et al., 2025). We refer the readers to this survey (Jha et al., 2022) for more details.

Under the existing NP setting, more members were introduced with different inductive biases in the model (Jha et al., 2022; Bruinsma et al., 2023; Dutordoir et al., 2023; Jung et al., 2024; Vadeboncoeur et al., 2023). For instance, attentive neural processes (Kim et al., 2019) incorporated dependencies between observations with attention neural works. Convolutional neural processes (Foong et al., 2020; Huang et al., 2023) assume translation equivariance among data points. These two methods explicitly defined the latent variable which requires density estimation. Recent works such as transformer neural processes (Nguyen & Grover, 2022) and neural diffusion processes (Dutordoir et al., 2023) turn to marginal likelihood maximization and do not have the latent distribution.

Other neural processes that claim to provide exact (Markou et al., 2022) or tractable inference (Lee et al., 2023; Wang et al., 2023) have been introduced. Stable neural processes (Liu et al., 2024) argued that NPs are prone to noisy context points and proposed a weighted likelihood model that focuses on subsets that are difficult to predict, but do not focus regularizing the posterior distribution. Compared to variational inference based methods, non-VI predictions can be less robust to noisy inputs in the data (Futami et al., 2018). It is also challenging to incorporate prior knowledge (Zhang et al., 2018) into these neural processes, which could be beneficial when no data is observed for the task. Several works introduced strategies to incorporate explicit priors in the function space rather than low-dimensional latent variables, we refer the readers to (Ma et al., 2019), (Ma & Hernández-Lobato, 2021), (Rodríguez-Santana et al., 2022) and (Rudner et al., 2022) for their formulations.

**Robust divergences.** Divergences in variational inference can be viewed as an regularization on the posterior distribution via the prior distribution. The commonly adopted KL divergence which minimizes the expected density ratio between the posterior and the prior is notorious for underestimating the true variance of the target distribution (Regli & Silva, 2018). Several other divergences have since been proposed to focus on obtaining a robust posterior when the input and output features are noisy or when there are outliers in the dataset.

Examples of robust divergences include Rényi divergence (Lee & Shin, 2022), beta and alpha divergences (Futami et al., 2018; Regli & Silva, 2018) which require additional parameters to control the density ratio so that the posterior can focus more on mass covering, mode seeking abilities or is robust against outliers based on prior knowledge. (Santana et al., 2022) utilized $\alpha$ divergence in implicit processes and observed robust prediction against model misspecification. $f$-divergence (Cheng et al., 2021; Wan et al., 2020) variational inferences provide a unification of different divergences under a general definition of a con-

vex function $f$ but would require the specification of the function $f$ as well as its dual function. Generalized variational inference (Knoblauch et al., 2019) suggested that any form of divergence can be used to replace the KL objective when the model is misspecified. As far as we understand, we are the first to analyse the limitations of NPs from the perspective of prior misspecifications, and facilitate robust divergence to enable better NP learning.

## 5. Experiments

**Datasets and training details.** We evaluate the proposed method on multiple regression tasks: 1D regression (Garnelo et al., 2018a; Gordon et al., 2019; Kim et al., 2019; Nguyen & Grover, 2022), image inpainting (Gordon et al., 2019; Nguyen & Grover, 2022). 1D regression includes three Gaussian Process (GP) regression tasks with different kernels: RBF, Matern 5/2 and Periodic. Image inpainting involves 2D regression on three image datasets: MNIST, SVHN and CelebA. Given some pixel coordinates $\mathbf{x}$ and intensities $\mathbf{y}$ as context, the goal is to predict the pixel value for the rest of image. More details about the training setups can be found in A.3.

**Baselines.** We first validate our approach on state-of-the-art NP families: neural processes (NP) (Garnelo et al., 2018b), attentive neural processes (ANP) (Kim et al., 2019), Bayesian aggregation neural processes (BA-NP) (Volpp et al., 2021), transformer neural processes with diagonal covariances (TNP-D) (Nguyen & Grover, 2022), and versatile neural processes (VNP) (Guo et al., 2023). For VNPs, they chose different parameterizations for the prior and posterior models, which can be used to validate if our objective is superior than simply decoupling the two models. We generalize the NP objective to RNP using Eq 12 or Eq 6 depending on whether the baseline model infer the latent distribution $p(\mathbf{z})$. The methods are considered as a special case of $\alpha = 1$ of RNP if the baseline model uses the VI objective or $\alpha = 0$ if the baseline model uses the ML-objective. The number of samples $K$ for the Monte Carlo is 32 for training and 50 for inference. Our experiments aim to answer the following research questions:

(1) How does RNPs perform under parameterization-related prior misspecification? To achieve this, We compare RNPs with SOTA NP frameworks on predictive performance. (2) How does RNPs perform under context-related prior misspecification? Here we introduce various types of in the context set such as domain shift and noisy context. (3) How to select the optimal $\alpha$ values? We also carry out ablation studies investigating how to select the optimal $\alpha$ values, the number of MC samples and the number of context points for our RNP framework.

### 5.1. Predictive performance under parameterization-related prior misspecification

Here we investigate prior misspecification caused by poor parameterization of $\varphi_{poor}$ in $q_\varphi(\mathbf{z}|\mathbb{C})$, which can happen in NPs because their posterior and prior models are forced to couple the parameters. We focus on the predictive performance, i.e., test log-likelihood, of both the context and target sets across different datasets. Specifically, we adopted the VI-based RNP objective to train NPs, ANPs and VNPs as their model designs include the prior models. We used the ML-based RNP objective to train TNP and BANP because the TNP objective was originally defined using ML only and the ML objective significantly outperformed the VI objective for BANPs. We set $\alpha = 0.7$ to train for VI-based RNPs and analogously $\alpha = 0.3$ for ML-based baselines. However, we show in section 5.3 that, in fact, these values can be set optimally via cross-validation. To put the baseline models in the spectrum, $\alpha = 1$ corresponds to the standard VI solutions (using the KLD), and $\alpha = 0$ corresponds to the maximum likelihood solutions.

Table 1 shows the mean test log-likelihood $\pm$ one standard deviation using 5 different random seeds for each method. We see that RNP consistently improved log-likelihood over the other two objectives and ranked the highest for all the baselines. RNP also consistently achieved better likelihood on TNP-D and VNP which generally outperform other baseline models across datasets. Some prominent improvements were achieved in harder tasks in 1D regression such as ANP Periodic and BA-NP Periodic where the vanilla NP objectives underperform. As previously illustrated in Fig 1(b) and Fig 1(c), RNP improves predictive performance by mitigating the oversmoothed predictions on periodic data. This could suggest that a misspecified prior model in the vanilla NP objective imposes an unjustifiable regularization on the posterior and hinder the expressiveness of the posterior and consequently predictive performance. RNP also significantly improved test likelihood of BA-NP and VNP on image inpainting tasks, demonstrating the superiority of RNPs on higher dimensional data.

### 5.2. Predictive performance under context-related prior misspecification

Here we designed another set of experiments where the prior model $q(\mathbf{z}|\mathbb{C})$ is clearly misspecified due to poor context data $\mathbb{C}_{poor}$. We first tested the framework under noisy contexts and then utilized domain shift datasets such as Lotka-Volterra and EMNIST (more detailed settings can found in section A.9). Supp Table 4 shows test log-likelihood under noisy context where we inject noises to the context labels. RNPs still outperformed baseline VI methods despite the general deterioration of context corruption.

Table 2 shows the test log-likelihood for the high-performing

*Table 1.* Test set log-likelihood ↑. The **bold** results indicate significant improvements of the RNP objective with p value $< 0.05$.

| Model | Set | Objective | RBF | Matern 5/2 | Periodic | MNIST | SVHN | CelebA | Avg rank |
|---|---|---|---|---|---|---|---|---|---|
| NP (Garnelo et al., 2018b) | context | $\mathcal{L}_{VI}$ | 0.69±0.01 | 0.56±0.02 | -0.49±0.01 | 0.99±0.01 | 3.24±0.02 | 1.71±0.04 | 2.3 |
| | | $\mathcal{L}_{ML}$ | 0.68±0.02 | 0.55±0.02 | -0.48±0.03 | 1.00±0.01 | 3.22±0.03 | 1.70±0.03 | 2.5 |
| | | $\mathcal{L}_{RNP(\alpha)}$ | **0.78±0.01** | **0.66±0.01** | -0.49±0.00 | 1.01±0.02 | 3.26±0.01 | 1.72±0.05 | 1.2 |
| | target | $\mathcal{L}_{VI}$ | 0.26±0.01 | 0.09±0.02 | -0.61±0.00 | 0.90±0.01 | 3.08±0.01 | 1.45±0.03 | 2.3 |
| | | $\mathcal{L}_{ML}$ | 0.28±0.02 | 0.11±0.02 | -0.61±0.01 | 0.92±0.01 | 3.07±0.02 | 1.47±0.02 | 1.8 |
| | | $\mathcal{L}_{RNP(\alpha)}$ | **0.33±0.01** | **0.16±0.01** | -0.62±0.00 | 0.91±0.01 | 3.09±0.01 | 1.45±0.01 | 1.7 |
| ANP (Kim et al., 2019) | context | $\mathcal{L}_{VI}$ | 1.38±0.00 | 1.38±0.00 | 0.65±0.04 | 1.38±0.00 | 4.14±0.00 | 3.92±0.07 | 1.3 |
| | | $\mathcal{L}_{ML}$ | 1.38±0.00 | 1.38±0.00 | 0.63±0.03 | 1.38±0.00 | 4.14±0.01 | 3.86±0.07 | 1.7 |
| | | $\mathcal{L}_{RNP(\alpha)}$ | 1.38±0.00 | 1.38±0.00 | **1.22±0.02** | 1.38±0.00 | 4.14±0.00 | 3.97±0.03 | 1.0 |
| | target | $\mathcal{L}_{VI}$ | 0.81±0.00 | 0.64±0.00 | -0.91±0.02 | 1.06±0.01 | 3.65±0.01 | 2.24±0.03 | 1.7 |
| | | $\mathcal{L}_{ML}$ | 0.80±0.00 | 0.64±0.00 | -0.89±0.02 | 1.04±0.01 | 3.65±0.01 | 2.23±0.03 | 2.2 |
| | | $\mathcal{L}_{RNP(\alpha)}$ | **0.84±0.00** | **0.67±0.00** | **-0.57±0.01** | 1.05±0.01 | 3.61±0.02 | 2.24±0.02 | 1.3 |
| BA-NP (Volpp et al., 2021) | context | $\mathcal{L}_{VI}$ | 1.43±0.03 | 1.04±0.08 | -0.65±0.02 | 0.81±0.84 | 2.76±0.59 | 1.65±0.01 | 2.3 |
| | | $\mathcal{L}_{ML}$ | 1.30±0.09 | 0.69±0.05 | -0.70±0.07 | 3.62±0.06 | 4.87±0.05 | 2.02±0.02 | 2.3 |
| | | $\mathcal{L}_{RNP(\alpha)}$ | 1.45±0.04 | 1.01±0.04 | **-0.41±0.02** | **3.85±0.09** | 4.88±0.05 | 2.00±0.02 | 1.3 |
| | target | $\mathcal{L}_{VI}$ | 1.19±0.03 | 0.79±0.09 | -0.89±0.01 | 0.24±0.64 | 2.60±0.50 | 1.30±0.01 | 2.5 |
| | | $\mathcal{L}_{ML}$ | 1.12±0.08 | 0.53±0.04 | -0.91±0.05 | 3.56±0.06 | 4.29±0.04 | 1.63±0.01 | 2.3 |
| | | $\mathcal{L}_{RNP(\alpha)}$ | 1.22±0.04 | 0.79±0.03 | **-0.72±0.02** | **3.79±0.09** | 4.31±0.02 | 1.61±0.02 | 1.2 |
| TNP-D (Nguyen & Grover, 2022) | context | $\mathcal{L}_{ML}$ | 2.58±0.01 | 2.57±0.01 | -0.52±0.00 | 1.73±0.11 | 10.63±0.12 | 4.61±0.27 | 1.8 |
| | | $\mathcal{L}_{RNP(\alpha)}$ | 2.59±0.00 | **2.59±0.00** | -0.52±0.00 | 1.81±0.12 | 10.72±0.08 | 4.66±0.23 | 1.0 |
| | target | $\mathcal{L}_{ML}$ | 1.38±0.01 | 1.03±0.00 | -0.59±0.00 | 1.63±0.07 | 6.69±0.04 | 2.45±0.05 | 1.8 |
| | | $\mathcal{L}_{RNP(\alpha)}$ | **1.41±0.00** | **1.04±0.00** | -0.59±0.00 | 1.67±0.07 | 6.71±0.04 | 2.46±0.06 | 1.0 |
| VNP (Guo et al., 2023) | context | $\mathcal{L}_{VI}$ | 1.37±0.00 | 1.37±0.00 | 1.23±0.03 | 1.60±0.10 | 0.80±0.00 | 0.08±0.03 | 2.0 |
| | | $\mathcal{L}_{RNP(\alpha)}$ | **1.38±0.00** | **1.38±0.00** | **1.32±0.01** | **3.63±0.39** | **4.00±0.06** | **2.65±0.06** | 1.0 |
| | target | $\mathcal{L}_{VI}$ | 0.90±0.02 | 0.70±0.03 | -0.49±0.00 | 1.59±0.10 | 0.80±0.00 | 0.08±0.03 | 2.0 |
| | | $\mathcal{L}_{RNP(\alpha)}$ | 0.92±0.01 | 0.71±0.03 | **-0.48±0.00** | **3.62±0.37** | **3.89±0.06** | **2.49±0.06** | 1.0 |

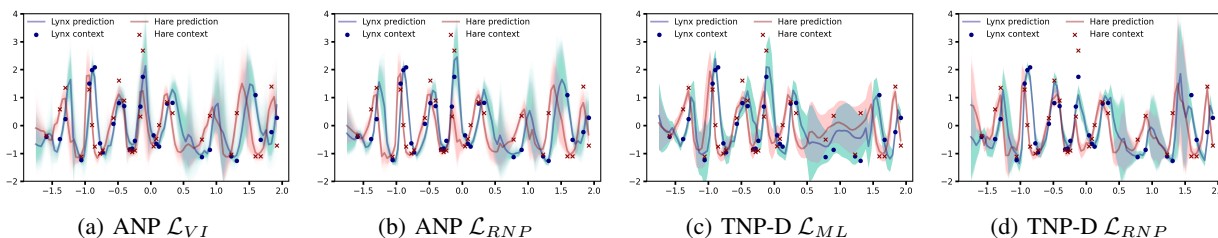

| (a) ANP $\mathcal{L}_{VI}$ | (b) ANP $\mathcal{L}_{RNP}$ | (c) TNP-D $\mathcal{L}_{ML}$ | (d) TNP-D $\mathcal{L}_{RNP}$ |
|---|---|---|---|

*Figure 2.* Prior misspecification experiment. Both models are trained on simulated Lotka-Volterra data and tested on the real-world Hare-Lynx dataset.

baseline TNP-D on two misspecified cases where tasks are generated from different distributions during the meta training and meta testing phase. For the 1D regression task, the model is trained using the Lotka-Volterra dataset which is generally used for prey-predator simulations. The dynamics is controlled by a two-variable ordinary differential equations: $\dot{x} = \theta_1 x - \theta_2 xy, \dot{y} = -\theta_3 y + \theta_4 xy$ where $x$ and $y$ correspond to the populations of the prey and predator respectively. The parameters are chosen as $\theta_1 = 1, \theta_2 = 0.01, \theta_3 = 0.5, \theta_4 = 0.01$ following (Gordon et al., 2019). The number of context points is randomly sampled $M \sim \mathcal{U}(15, 100)$, and the number of target points is $N \sim \mathcal{U}(15, 100 - M)$. We choose 20,000 functions for training, and sample another 1,000 functions for evaluation.

We then test the model on a real-world Hare-Lynx dataset which tracks the two species populations over 90 years. The input and output features were normalized via z-score normalization. Our method in table 4 shows outperformance

across multiple datasets as the impact of misspecified contexts is alleviated via the divergence. The results in table 2 show that RNP significantly outperformed the ML objective on both the training and testing data, highlighting the robustness of our objective. Fig 2 shows the prediction results on the Lynx dataset, where the RNP achieves better uncertainty estimate and tracks the seasonality of the data more efficiently than the ML objective. We also tested TND-D on the Extended MNIST dataset with 47 classes that include letters and digits. We use classes 0-10 for meta training and hold out classes 11-46 for meta testing under prior misspecification. Table 2 shows that RNP performed slightly worse on the EMNIST training task but significantly outperformed the ML objective on the test set (last column), which demonstrates the superior robustness of the new objective under misspecification.

*Table 2.* Loglikelihood ↑ under prior misspecification using TNP-D. The **bold** results indicate significant improvements with p<0.05.

| Objective | $\mathbb{D}_{\text{train}}$ (Lotka-Volterra) | | Misspec $\mathbb{D}_{\text{test}}$ (Hare-Lynx) | | $\mathbb{D}_{\text{train}}$ EMNIST (class 0-10) | | Misspec $\mathbb{D}_{\text{test}}$ (class 11-46) | |
|---|---|---|---|---|---|---|---|---|
| | context | target | context | target | context | target | context | target |
| $\mathcal{L}_{ML}$ | 3.09±0.22 | 1.98±0.11 | -0.59±0.47 | -4.44±0.41 | 1.54±0.05 | 1.56±0.07 | 0.03±0.97 | -0.20±0.57 |
| $\mathcal{L}_{RNP}$ | 3.32±0.15 | **2.12±0.06** | -0.17±0.31 | **-3.63±0.09** | 1.52±0.08 | 1.47±0.12 | 0.96±0.18 | **0.70±0.15** |

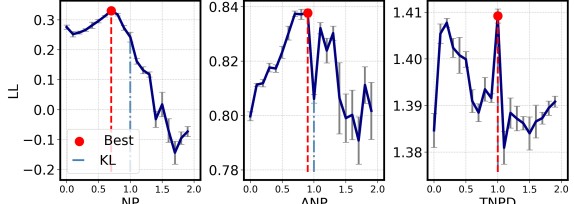

(a) $\alpha$ on RBF. The red line indicates the best results, and the blue line corresponds to the KL results.

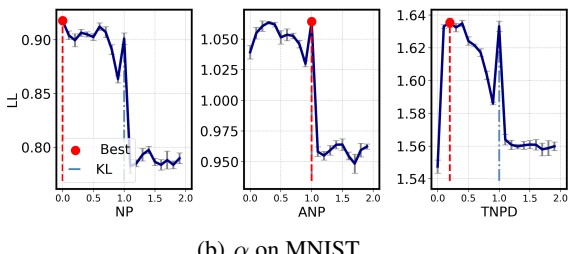

(b) $\alpha$ on MNIST.

*Figure 3.* Hyperparameter($\alpha$) tuning. cross-validation is used to select the optimal $\alpha \in (0, 2)$.

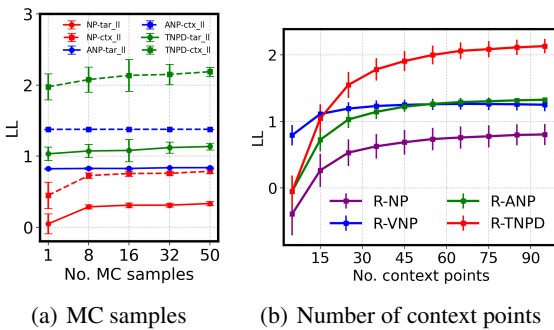

(a) MC samples  (b) Number of context points

*Figure 4.* Ablation study. We investigated how MC sample sizes and the number of context points affect test log-likelihood.

3. Gradually decrease $\alpha$ (with granularity according to computational constraints) to 0. The intuition is inspired by KL annealing for VAE models (Bowman et al., 2016), which starts with a strong prior penalization and gradually reduces the prior penalization and focuses more on model expressivity. Our results in Supp Table 5 show that this automatic $\alpha$ tuning strategy still outperformed the baselines.

### 5.4. Ablation studies

**Effects of Monte Carlo samples on likelihood.** As both RNP (Eq 6) and RNP-ML (Eq 12) require MC approximations, we investigate the effects of the number of MC samples $K$ on predictive likelihood. We set $K \in \{1, 8, 16, 32, 50\}$ for optimizing the RNP objective during training and use $K = 50$ for inference. Note that $K = 1$ corresponds to the deterministic NPs (conditional NPs).

Fig 4(a) shows both the context and target log-likelihood for three methods: NP, ANP and TNP-D on the RBF dataset. As expected, increasing the number of MC samples improves the LL mean and also reduces the variance for all the methods with $K = 50$ achieving the highest LL and the smallest variance. In practice, we set $K = 32$ to balance performance and memory efficiency. However, it takes as few as 8 samples during training to gain better estimates, and one can choose to increase the number of samples during inference for better predictive performance or to reduce it for scalability and real-time applications. We also reported the wall clock time between standard NPs and RNPs in Supp Table 7 and no significant differences were found with our objective. Our computational complexity is linear to the number of MC samples, which is also comparable to the VI objective.

### 5.3. How to select the optimal $\alpha$ values?

We have shown that choosing $\alpha = 0.7$ for the VI objectives and $\alpha = 0.3$ for the ML objectives provides significant performance improvements over the competing approaches. One may also use the prior knowledge to select the $\alpha$ based on the understanding of the misspecification. Nevertheless, we recognize that in other scenarios such as very different datasets and/or models, this default value may not work as well as in our experiments. For this purpose, we have found that cross-validation is an effective tool for finding near-optimal values (Futami et al., 2018). We hypersearched the $\alpha$ values from 0 to 2 with an interval of 0.1. As shown in Fig 3(a) and Fig 3(b), the optimal solutions are model and dataset specific. Similarly, (Rodríguez-Santana & Hernández-Lobato, 2022) tuned $\alpha$ for general Bayesian inference and showed that the selection of $\alpha$ is relevant to different error metrics. Lastly, since cross-validation is computationally expensive, we adopted the following heuristics:

1. start with a value close to 1, which corresponds to standard KL minimization.

2. Only consider $0 < \alpha < 1$ since otherwise we may violate the conditions for the divergence between two Gaussians.

**Effects of the number of context points on likelihood.** We study the effect of the context points on the target likelihood. During training the number of context points is sampled from $\mathcal{U}(3, 50)$ and vary the number of context points from 5 to 95 at an interval of 10 for evaluation. The results of RBF in Fig 4(b) shows that increasing the context set size leads to improved LL for all the methods. Most methods (e.g., NP, ANP, VNP) plateaued after the number of contexts increases to more than 45, whereas TNP-D still shows unsaturated performance improvement with the increased context size.

**Different prior and posterior parameterization.** To validate if a simple strategy of decoupling the prior and posterior coupling works for NPs, we compared the RNP with the separate prior-posterior parameterization for NPs and ANPs. The results in Supp Table 6 showed that RNP still outperformed this baseline with much fewer parameters. The theoretical justification of the baseline could be the violation of the consistency property of NPs where different parameterizations of $\phi$ and $\varphi$ make the KL term non-zero and the marginal is no longer consistent.

## 6. Conclusion

We have proposed the Rényi Neural Process (RNP), a new NP framework designed to mitigate prior misspecification in neural processes. RNPs bridge the variational inference and maximum likelihood estimation objectives in vanilla NPs through the use of the Rényi divergence. We have shown the superiority of our generalized objective in improving predictive performance by selecting optimal $\alpha$ values. We have applied our framework to multiple state-of-the-art NP models and observed consistent log-likelihood improvements across benchmarks, including 1D regression, image inpainting, and real-world regression tasks. A limitation of our framework lies in drawing multiple samples with Monte-Carlo, which sacrifices some computational efficiency in exchange for better predictive performance, due to the infeasibility of computing analytical solutions on the divergence. To further improve our efficiency, we suggest to adopt variance reduction methods such as double-reparameterized gradient estimator that could require fewer MC samples. Additionally, for attention-based NPs, the computational complexity of self-attention is $O(N^2)$, which impedes real-time applications. We propose to adopt efficient attention mechanisms, e.g., Nyströmformer which uses low rank approximation of the attention matrix and has the complexity of $O(N)$.

## Acknowledgements

This work was supported by resources and expertise provided by CSIRO IMT Scientific Computing. We would like to thank our colleague Rafael Oliveira from CSIRO's Data 61 and Kian Ming Chai from DSO National Laboratories for their valuable insights on the theory discussions. We would like to thank the anonymous reviewers for the constructive feedback and advice.

## Impact Statement

The social impact of our work is to improve the robustness of generative models and to enable reliable machine learning algorithms. Negative impacts of our model include recovering facial photos from incomplete images for fraudulent scams.

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

# A. Appendix

## A.1. Pseudocode

---

**Algorithm 1** Rényi Neural Processes

---

    **Input**: Context inputs $X_C$ and outputs $Y_C$. Target inputs $X_T$, and target outputs $Y_T$ during training
    **Output**: Target distribution $p(Y_T|X_T, X_C, Y_C)$
    **Training:**
    **for** epoch=1 **to** max_epoch **do**
        Sample a context set $(X_C, Y_C)$ and a target set $(X_T, Y_T)$
        Obtain the posterior distribution with the encoding network: $q_\varphi(\mathbf{z}|X_T, Y_T, X_C, Y_C)$
        Obtain the approximated prior distribution with the encoding network: $q_\varphi(\mathbf{z}|X_C, Y_C)$
        Sample $\mathbf{z}_1, ..., \mathbf{z}_K \sim q_\varphi(\mathbf{z}|X_T, Y_T, X_C, Y_C)$
        Construct the likelihood model with the decoding network: $p_\theta(Y_T|X_T, \mathbf{z}_k)$
        Compute the objective $\mathcal{L}_{RNP}$ using Eq 6 or Eq 12
        Update the encoder parameters with $\nabla_\varphi \mathcal{L}$ (Eq 8) and the decoder parameters with $\nabla_\theta \mathcal{L}$
    **end for**
    **Inference:**
    Construct the approximated prior $q_\varphi(\mathbf{z}|X_C, Y_C)$
    Sample $\mathbf{z}_1, ..., \mathbf{z}_K \sim q_\varphi(\mathbf{z}|X_C, Y_C)$
    Predict the target distribution $p_\theta(Y_T|X_T, \mathbf{z}_k)$
    Estimate the log-likelihood of the target outputs using Eq 13.

---

## A.2. Notation

The notations used in the article is summarized in Table 3.

*Table 3.* Notation

| Name | Description |
|------|-------------|
| $f$ | function sample from a stochastic process |
| $\mathcal{X}$ | input space |
| $\mathcal{Y}$ | output space |
| $\mathcal{Z}$ | latent space |
| $\mathbf{x}$ | input features |
| $\mathbf{y}$ | output features |
| $\mathbf{z}$ | latent variable representing stochasticity of the functional sample $f$ |
| $X_C$ | $\mathbb{R}^{M \times D}$ context inputs |
| $Y_C$ | $\mathbb{R}^{M \times 1}$ context outputs |
| $X_T$ | $\mathbb{R}^{N \times D}$ target inputs |
| $Y_T$ | $\mathbb{R}^{N \times 1}$ target outputs |
| $M$ | number of samples in the context set, indexed by $m$ |
| $N$ | number of samples in the target set, indexed by $n$ |
| $D_x$ | dimension of input features |
| $D_y$ | dimension of input features |
| $\mathbb{C}, \mathbb{T}$ | notation for Context and Target |
| $\varphi$ | parameters in the posterior model $q(\mathbf{z}|X, Y)$ |
| $\theta$ | parameters in the likelihood model $p(Y|\mathbf{z}, X)$ |
| $K$ | number of $\mathbf{z}$ samples for Monte Carlo approximation |

### A.3. Training details

For 1D regression tasks, given a function $f$ sampled from a GP prior with varying scale and length and a context set generated by such function, our goal is to predict the target distribution. The number of context points is randomly sampled $M \sim \mathcal{U}(3, 50)$, and the number of target points is $N \sim \mathcal{U}(3, 50 - M)$ (Nguyen & Grover, 2022). We choose 100,000 functions for training, and sample another 3,000 functions for testing. The input features were normalized to $[-2, 2]$.

Image inpainting involves 2D regression on three image datasets: MNIST, SVHN and CelebA. Given some pixel coordinates $\mathbf{x}$ and intensities $\mathbf{y}$ as context, the goal is to predict the pixel value for the rest of image. The number of context points for inpainting tasks is $M \sim \mathcal{U}(3, 200)$ and the target point count is $N \sim \mathcal{U}(3, 200 - M)$. The input coordinates were normalized to $[-1, 1]$ and pixel intensities were rescaled to $[-0.5, 0.5]$. All the models can be trained using a single GPU with 16GB memory.

### A.4. Proof of Proposition 3.2

The true ELBO of neural processes without prior approximation can be written as:

$$ELBO = \mathbb{E}_{q_\varphi(\mathbf{z}|X_T, Y_T, X_C, Y_C)} \log p_\theta(Y_T|X_T, \mathbf{z}) - D_{\mathrm{KL}}\left(q_\varphi(\mathbf{z}|X_T, Y_T, X_C, Y_C) \| p(\mathbf{z}|X_C, Y_C)\right) \tag{14a}$$

$$= \mathbb{E}_{q_\varphi(\mathbf{z}|\mathbb{C}, \mathbb{T})}[\log p_\theta(Y_T|X_T, \mathbf{z}) + \log p(\mathbf{z}|\mathbb{C}) - \log q_\varphi(\mathbf{z}|\mathbb{C}, \mathbb{T}) + \log q_\varphi(\mathbf{z}|\mathbb{C}) - \log q_\varphi(\mathbf{z}|\mathbb{C})] \tag{14b}$$

$$= \mathbb{E}_{q_\varphi(\mathbf{z}|\mathbb{C}, \mathbb{T})}\left[\log \frac{p_\theta(Y_T|X_T, \mathbf{z})q_\varphi(\mathbf{z}|\mathbb{C})}{q_\varphi(\mathbf{z}|\mathbb{C}, \mathbb{T})} + \log \frac{p(\mathbf{z}|\mathbb{C})}{q_\varphi(\mathbf{z}|\mathbb{C})}\right] \tag{14c}$$

$$= \mathbb{E}_{q_\varphi(\mathbf{z}|\mathbb{C}, \mathbb{T})}\left[\log \frac{p_\theta(Y_T|X_T, \mathbf{z})q_\varphi(\mathbf{z}|\mathbb{C})}{q_\varphi(\mathbf{z}|\mathbb{C}, \mathbb{T})}\right] + \mathbb{E}_{q_\varphi(\mathbf{z}|\mathbb{C}, \mathbb{T})}\left[\log \frac{p(\mathbf{z}|\mathbb{C})}{q_\varphi(\mathbf{z}|\mathbb{C})}\right] \tag{14d}$$

$$= -\mathcal{L}_{VI} + \mathbb{E}_{q_\varphi(\mathbf{z}|\mathbb{C}, \mathbb{T})} \log \frac{p(\mathbf{z}|\mathbb{C})}{q_\varphi(\mathbf{z}|\mathbb{C})} \tag{14e}$$

Therefore, the VI objective will only recover the true ELBO only when $q_\varphi(\mathbf{z}|\mathbb{C}) = p(\mathbf{z}|\mathbb{C})$, i.e., the ground truth posterior is identified and the prior model is well-specified. When $q_\varphi(\mathbf{z}|\mathbb{C}) \neq p(\mathbf{z}|\mathbb{C})$, which is usually the case, the model is optimizing an objective which is not necessarily the true ELBO.

Furthermore, we will derive the gradients of the true ELBO and the approximate ELBO $\mathcal{L}_{VI}$ w.r.t the parameters of the posteriors using the re-parameterization trick:

$$\nabla_\varphi ELBO = \nabla_\varphi \left(\mathbb{E}_{q_\varphi(\mathbf{z}|\mathbb{C}, \mathbb{T})}\left[\log \frac{p_\theta(Y_T|X_T, \mathbf{z})p(\mathbf{z}|\mathbb{C})}{q_\varphi(\mathbf{z}|\mathbb{C}, \mathbb{T})}\right]\right) \tag{15a}$$

$$= \mathbb{E}_\epsilon \nabla_\varphi \left[\log p(Y_T, |X_T, g_\varphi(\epsilon)) + \log p(g_\varphi(\epsilon)|\mathbb{C}) - \log q_\varphi(g_\varphi(\epsilon))\right], \quad g_\varphi(\epsilon) = g(\epsilon, \varphi, \mathbb{C}, \mathbb{T}) \tag{15b}$$

$$= \mathbb{E}_\epsilon \left[\nabla_\varphi \log p(Y_T, |X_T, g_\varphi(\epsilon)) + \nabla_\varphi \log p(g_\varphi(\epsilon)|\mathbb{C}) - \nabla_\varphi \log q_\varphi(g_\varphi(\epsilon))\right] \tag{15c}$$

As we do not know the form of conditional prior $p(g_\varphi(\epsilon)|\mathbb{C})$ in Eq 15, the gradient is difficult to estimate. Therefore, the VI provides a way to approximate the gradient with

$$\nabla_\varphi \mathcal{L}_{VI} = \nabla_\varphi \left(\mathbb{E}_{q_\varphi(\mathbf{z}|\mathbb{C}, \mathbb{T})}\left[\log \frac{p_\theta(Y_T|X_T, \mathbf{z})q_\varphi(\mathbf{z}|\mathbb{C})}{q_\varphi(\mathbf{z}|\mathbb{C}, \mathbb{T})}\right]\right) \tag{16a}$$

$$= \mathbb{E}_\epsilon \left[\nabla_\varphi \log p(Y_T, |X_T, g_\varphi(\epsilon)) + \nabla_\varphi \log q_\varphi(g_\varphi(\epsilon)|\mathbb{C}) - \nabla_\varphi \log q_\varphi(g_\varphi(\epsilon))\right] \tag{16b}$$

Hence, now we have a "learned" prior model $q_\varphi(\mathbf{z}|\mathbb{C})$ that can move towards posterior samples $\mathbf{z} \sim q_\varphi(\mathbf{z}|\mathbb{C}, \mathbb{T})$. However, as there is no guidance to regularize $q_\varphi(\mathbf{z}|\mathbb{C})$ to be close to the true prior, this prior approximation can be misspecified, which can inevitably cause a biased estimate of the posterior.

## A.5. Derivation of $\mathcal{L}_{RNP}$ (eq 6)

$$\min_{\theta,\varphi} D_\alpha \left( q_\varphi(\mathbf{z}|X_T, Y_T, X_C, Y_C) \| p(\mathbf{z}|X_T, Y_T, X_C, Y_C) \right) \tag{17}$$

$$= \min_{\theta,\varphi} \frac{1}{\alpha - 1} \log \mathbb{E}_{q_\varphi(\mathbf{z}|X_T,Y_T,X_C,Y_C)} \left[ \frac{p(\mathbf{z}|X_T, Y_T, X_C, Y_C)}{q_\varphi(\mathbf{z}|X_T, Y_T, X_C, Y_C)} \right]^{1-\alpha} \text{(By definition)} \tag{18}$$

$$= \max_{\theta,\varphi} \frac{1}{1 - \alpha} \log \mathbb{E}_{q_\varphi(\mathbf{z}|X_T,Y_T,X_C,Y_C)} \left[ \frac{p(\mathbf{z}, Y_T|X_T, X_C, Y_C)}{q_\varphi(\mathbf{z}|X_T, Y_T, X_C, Y_C)} \right]^{1-\alpha} + Const.\text{(Split marginal likelihood)} \tag{19}$$

$$= \max_{\theta,\varphi} \frac{1}{1 - \alpha} \log \mathbb{E}_{q_\varphi(\mathbf{z}|X_T,Y_T,X_C,Y_C)} \left[ \frac{p(\mathbf{z}, Y_T|X_T, X_C, Y_C)}{q_\varphi(\mathbf{z}|X_T, Y_T, X_C, Y_C)} \right]^{1-\alpha} \text{(Equivalence of removing the constant)} \tag{20}$$

$$\approx \max_{\theta,\varphi} \frac{1}{1 - \alpha} \log \mathbb{E}_{q_\varphi(\mathbf{z}|X_T,Y_T,X_C,Y_C)} \left[ \frac{p_\theta(Y_T|X_T, \mathbf{z}) q_\varphi(\mathbf{z}|X_C, Y_C)}{q_\varphi(\mathbf{z}|X_T, Y_T, X_C, Y_C)} \right]^{1-\alpha} \tag{21}$$

## A.6. Derivation of $\mathcal{L}_{RNPML}$ (eq 12)

We start by rewriting the ML objective (Eq 9) as minimizing the KL divergence:

$$- \mathcal{L}_{ML}(\theta) = \max_\theta \mathbb{E}_{\mathbb{D}_{\text{train}}} \log p_\theta(Y_T|X_T, \mathbb{C}) \tag{22}$$

$$= \max_\theta \mathbb{E}_{\mathbb{D}_{\text{train}}} \left[ \frac{1}{N} \sum_{n=1}^{N} \log p_\theta(\mathbf{y}_n|\mathbf{x}_n, \mathbb{C}) \right] + Const \quad \text{(Average likelihood for stabilized training)} \tag{23}$$

$$\approx \max_\theta \mathbb{E}_{\mathbb{D}_{\text{train}}} \int \hat{p}(\mathbf{y}|\mathbf{x}, \mathbb{C}) \log p_\theta(\mathbf{y}|\mathbf{x}, \mathbb{C}) d\mathbf{y} \text{( Definition of the empirical distribution)} \tag{24}$$

$$= \max_\theta \mathbb{E}_{\mathbb{D}_{\text{train}}} \int \hat{p}(\mathbf{y}|\mathbf{x}, \mathbb{C}) \left[ \log p_\theta(\mathbf{y}|\mathbf{x}, \mathbb{C}) - \log \hat{p}(\mathbf{y}|\mathbf{x}, \mathbb{C}) + \log \hat{p}(\mathbf{y}|\mathbf{x}, \mathbb{C}) \right] d\mathbf{y} \tag{25}$$

$$\equiv \min_\theta \mathbb{E}_{\mathbb{D}_{\text{train}}} \left[ D_{\text{KL}} \left( \hat{p}(\mathbf{y}|\mathbf{x}, \mathbb{C}) \| p_\theta(\mathbf{y}|\mathbf{x}, \mathbb{C}) \right) \right] \text{(Definition of KLD and removing the constant without } \theta) \tag{26}$$

We now replace the KLD with RD

$$\min_\theta \mathbb{E}_{\mathbb{D}_{\text{train}}} \left[ D_\alpha(\hat{p}(\mathbf{y}|\mathbf{x}, \mathbb{C}) \| p_\theta(\mathbf{y}|\mathbf{x}, \mathbb{C})) \right] \tag{27}$$

$$= \min_\theta \mathbb{E}_{\mathbb{D}_{\text{train}}} \frac{1}{\alpha - 1} \left[ \log \int \hat{p}^\alpha(\mathbf{y}|\mathbf{x}, \mathbb{C}) p_\theta^{1-\alpha}(\mathbf{y}|\mathbf{x}, \mathbb{C}) d\mathbf{y} \right] \text{(Definition of RD)} \tag{28}$$

$$\approx \min_\theta \mathbb{E}_{\mathbb{D}_{\text{train}}} \frac{1}{\alpha - 1} \left[ \log \sum_{n=1}^{N} (\frac{1}{N})^\alpha p_\theta^{1-\alpha}(\mathbf{y}_n|\mathbf{x}_n, \mathbb{C}) \right] \text{(Definition of the empirical distribution)} \tag{29}$$

$$= \min_\theta \mathbb{E}_{\mathbb{D}_{\text{train}}} \frac{1}{\alpha - 1} \log \sum_{n=1}^{N} p_\theta^{1-\alpha}(\mathbf{y}_n|\mathbf{x}_n, \mathbb{C}) + Const \text{( Split the non-}\theta \text{ term)} \tag{30}$$

$$\equiv \min_\theta \mathbb{E}_{\mathbb{D}_{\text{train}}} \frac{1}{(\alpha - 1)N} \log \sum_{n=1}^{N} p_\theta^{1-\alpha}(\mathbf{y}_n|\mathbf{x}_n, \mathbb{C}) \text{(Average likelihood for stabilized training)} \tag{31}$$

$$= \min_\theta \mathbb{E}_{\mathbb{D}_{\text{train}}} \frac{1}{(\alpha - 1)N} \sum_{n=1}^{N} \log \left( \int p_\theta(\mathbf{y}_n, \mathbf{z}|\mathbf{x}_n, \mathbb{C}) d\mathbf{z} \right)^{1-\alpha} \tag{32}$$

## A.7. Theoretical relationships between the $\mathcal{L}_{VI}$, $\mathcal{L}_{ML}$ and $\mathcal{L}_{RNP}$ objectives. 3.4

Our Rényi objective unifies the common three objectives for NPs: $\mathcal{L}_{VI}, \mathcal{L}_{ML}$ (maximum likelihood estimation), and $\mathcal{L}_{CNP}$ (conditional NPs or deterministic NPs).

$$- \mathcal{L}_{RNP} : \frac{1}{1-\alpha} \log \mathbb{E}_{q_\varphi(\mathbf{z}|X_T,Y_T,X_C,Y_C)} \left[ \frac{p_\theta(Y_T|X_T,\mathbf{z})q_\varphi(\mathbf{z}|X_C,Y_C)}{q_\varphi(\mathbf{z}|X_T,Y_T,X_C,Y_C)} \right]^{1-\alpha} \tag{33a}$$

$$- \mathcal{L}_{VI}(\alpha \to 1) : \quad \mathbb{E}_{q_\varphi(\mathbf{z}|X_T,Y_T,X_C,Y_C)} \log \left[ \frac{p_\theta(Y_T|X_T,\mathbf{z})q_\varphi(\mathbf{z}|X_C,Y_C)}{q_\varphi(\mathbf{z}|X_T,Y_T,X_C,Y_C)} \right] \tag{33b}$$

$$- \mathcal{L}_{ML}(\alpha = 0) : \log \mathbb{E}_{q_\varphi(\mathbf{z}|X_C,Y_C)} p_\theta(Y_T|X_T,\mathbf{z}) \tag{33c}$$

$$- \mathcal{L}_{CNP} \ (\alpha = 0 \text{ and } q_\varphi(\mathbf{z}|X_C,Y_C) = \delta(\varphi(X_C,Y_C)) : \log p_\theta(Y_T|X_T,\varphi(X_C,Y_C)) \tag{33d}$$

Proof of the ML objective:

$$\mathcal{L}_{RNP(\alpha=0)} = \log \mathbb{E}_{q_\varphi(\mathbf{z}|X_T,Y_T,X_C,Y_C)} \left[ \frac{p_\theta(Y_T|X_T,\mathbf{z})q_\varphi(\mathbf{z}|X_C,Y_C)}{q_\varphi(\mathbf{z}|X_T,Y_T,X_C,Y_C)} \right] \tag{34a}$$

$$= \log \int q_\varphi(\mathbf{z}|X_T,Y_T,X_C,Y_C) \left[ \frac{p_\theta(Y_T|X_T,\mathbf{z})q_\varphi(\mathbf{z}|X_C,Y_C)}{q_\varphi(\mathbf{z}|X_T,Y_T,X_C,Y_C)} \right] \tag{34b}$$

$$= \log \int p_\theta(Y_T|X_T,\mathbf{z})q_\varphi(\mathbf{z}|X_C,Y_C) = \log \mathbb{E}_{q_\varphi(\mathbf{z}|X_C,Y_C)} p_\theta(Y_T|X_T,\mathbf{z}) = \mathcal{L}_{ML} \tag{34c}$$

Proof of the NPVI objective: Next we will prove that $\mathcal{L}_{RNP,\alpha \to 1} = \mathcal{L}_{VI}$. Applying Theorem 5 from (Van Erven & Harremos, 2014a) to the new posteior and prior, we have:

$$\mathcal{D}_{\alpha \to 1} \left( q_\varphi(\mathbf{z}|X_T,Y_T,X_C,Y_C) || p(\mathbf{z}|X_T,Y_T,X_C,Y_C) \right) \tag{35a}$$

$$= \mathcal{KL} \left( q_\varphi(\mathbf{z}|X_T,Y_T,X_C,Y_C) || p(\mathbf{z}|X_T,Y_T,X_C,Y_C) \right) \tag{35b}$$

$$= \mathbb{E}_{q_\varphi(\mathbf{z}|X_T,Y_T,X_C,Y_C)} \log \frac{q_\varphi(\mathbf{z}|X_T,Y_T,X_C,Y_C)p(Y_T|X_C,Y_C,X_T)}{p(\mathbf{z},Y_T|X_T,X_C,Y_C)} \tag{35c}$$

$$= -\mathbb{E}_{q_\varphi(\mathbf{z}|X_T,Y_T,X_C,Y_C)} \log \frac{p(\mathbf{z},Y_T|X_T,X_C,Y_C)}{q_\varphi(\mathbf{z}|X_T,Y_T,X_C,Y_C)} + Const \tag{35d}$$

$$\equiv -\mathbb{E}_{q_\varphi(\mathbf{z}|X_T,Y_T,X_C,Y_C)} \left[ \log p(Y_T|\mathbf{z},X_T) + \log p_\varphi(\mathbf{z}|X_C,Y_C) - \log q_\varphi(\mathbf{z}|X_T,Y_T,X_C,Y_C) \right] = \mathcal{L}_{VI} \tag{35e}$$

## A.8. Gradients of Objectives for Rényi Neural Processes

$$\nabla \varphi \mathcal{L}_{RNP} = \frac{1}{1-\alpha} \nabla \varphi \log \mathbb{E}_{q_\varphi(\mathbf{z}|\mathbb{C},\mathbb{T})} \left[ \frac{p_\theta(Y_T|X_T,\mathbf{z})q_\varphi(\mathbf{z}|\mathbb{C})}{q_\varphi(\mathbf{z}|\mathbb{C},\mathbb{T})} \right]^{1-\alpha} \tag{36a}$$

$$= \frac{1}{1-\alpha} \left( \mathbb{E}_{q_\varphi(\mathbf{z}|\mathbb{C},\mathbb{T})} \left[ \frac{p_\theta(Y_T|X_T,\mathbf{z})q_\varphi(\mathbf{z}|\mathbb{C})}{q_\varphi(\mathbf{z}|\mathbb{C},\mathbb{T})} \right]^{1-\alpha} \right)^{-1} \mathbb{E}_{q_\varphi(\mathbf{z}|\mathbb{C},\mathbb{T})} \left( \nabla \varphi \left[ \frac{p_\theta(Y_T|X_T,\mathbf{z})q_\varphi(\mathbf{z}|\mathbb{C})}{q_\varphi(\mathbf{z}|\mathbb{C},\mathbb{T})} \right]^{1-\alpha} \right) \tag{36b}$$

$$= \frac{1}{1-\alpha} \left( \mathbb{E}_{q_\varphi(\mathbf{z}|\mathbb{C},\mathbb{T})} \left[ \frac{p_\theta(Y_T|X_T,\mathbf{z})q_\varphi(\mathbf{z}|\mathbb{C})}{q_\varphi(\mathbf{z}|\mathbb{C},\mathbb{T})} \right]^{1-\alpha} \right)^{-1} \mathbb{E}_{q_\varphi(\mathbf{z}|\mathbb{C},\mathbb{T})} \left( \left[ \frac{p_\theta(Y_T|X_T,\mathbf{z})q_\varphi(\mathbf{z}|\mathbb{C})}{q_\varphi(\mathbf{z}|\mathbb{C},\mathbb{T})} \right]^{1-\alpha} \nabla \varphi (1-\alpha) \log \frac{p_\theta(Y_T|X_T,\mathbf{z})q_\varphi(\mathbf{z}|\mathbb{C})}{q_\varphi(\mathbf{z}|\mathbb{C},\mathbb{T})} \right) \tag{36c}$$

$$= \mathbb{E}_{q(\mathbf{z}|\mathbb{C},\mathbb{T})} \left( w \nabla \varphi \log \frac{p_\theta(Y_T|X_T,\mathbf{z})q_\varphi(\mathbf{z}|\mathbb{C})}{q_\varphi(\mathbf{z}|\mathbb{C},\mathbb{T})} \right), \tag{36d}$$

$$w = \left[ \frac{p_\theta(Y_T|X_T,\mathbf{z})q_\varphi(\mathbf{z}|\mathbb{C})}{q_\varphi(\mathbf{z}|\mathbb{C},\mathbb{T})} \right]^{1-\alpha} / \left( \mathbb{E}_{q_\varphi(\mathbf{z}|\mathbb{C},\mathbb{T})} \left[ \frac{p_\theta(Y_T|X_T,\mathbf{z})q_\varphi(\mathbf{z}|\mathbb{C})}{q_\varphi(\mathbf{z}|\mathbb{C},\mathbb{T})} \right]^{1-\alpha} \right) \tag{36e}$$

Suppose $w_k = \frac{p_\theta(Y_T|X_T,\mathbf{z}_k)q_\varphi(\mathbf{z}_k|\mathbb{C})}{q_\varphi(\mathbf{z}_k|\mathbb{C},\mathbb{T})}$ where $\mathbf{z}_k \sim q_\varphi(\mathbf{z}|\mathbb{C},\mathbb{T})$. We have $\nabla \varphi \mathcal{L}_{RNP} = \sum_{k=1}^K \left( \frac{w_k^{1-\alpha}}{\sum_{k=1}^K w_k^{1-\alpha}} \nabla_\varphi \log w_k \right)$

In a simple case of a Bayesian linear regression model $y = \theta^T \mathbf{x} + \epsilon$ with two dimensional inputs and one dimensional output, the analytical solutions of the posterior is a mutilvariate Gaussian $p(\theta|\mathbf{x}, y) \sim \mathcal{N}(\mu, \Sigma)$. Suppose we use a factorized

*Table 4.* Test log-likelihood with noisy contexts.

| Model | Obj | RBF | Matern 5/2 | Periodic | MNIST | SVHN | CelebA |
|---|---|---|---|---|---|---|---|
| NP | $\mathcal{L}_{VI}$ | -0.53 $\pm$ 0.01 | -0.56 $\pm$ 0.01 | -0.74 $\pm$ 0.00 | 0.76 $\pm$ 0.02 | 2.81 $\pm$ 0.04 | 0.89 $\pm$ 0.07 |
| | $\mathcal{L}_{RNP}$ | **-0.45 $\pm$ 0.01** | **-0.50 $\pm$ 0.01** | **-0.73 $\pm$ 0.00** | **0.83 $\pm$ 0.02** | **2.98 $\pm$ 0.01** | **1.16 $\pm$ 0.02** |
| ANP | $\mathcal{L}_{VI}$ | -2.43 $\pm$ 0.19 | -2.15 $\pm$ 0.18 | -0.99 $\pm$ 0.01 | 0.90 $\pm$ 0.02 | 2.83 $\pm$ 0.06 | 1.55 $\pm$ 0.05 |
| | $\mathcal{L}_{RNP}$ | **-2.29 $\pm$ 0.12** | **-2.11 $\pm$ 0.15** | -1.20 $\pm$ 0.03 | **0.96 $\pm$ 0.01** | **3.11 $\pm$ 0.02** | **1.82 $\pm$ 0.03** |

*Table 5.* RNP results with automatic tuning of $\alpha$ values

| Model | Set | Setting | RBF | Matern 5/2 | Periodic | MNIST | SVHN |
|---|---|---|---|---|---|---|---|
| NP | context | $\mathcal{L}_{VI}$ | 0.69±0.01 | 0.56±0.02 | -0.49±0.01 | 0.99±0.01 | 3.24±0.02 |
| | | $\mathcal{L}_{RNP\_ada\alpha}$ | **0.75±0.02** | **0.61±0.02** | **-0.49±0.00** | **1.01±0.01** | **3.26±0.01** |
| | target | $\mathcal{L}_{VI}$ | 0.26±0.01 | 0.09±0.02 | **-0.61±0.00** | 0.90±0.01 | 3.08±0.01 |
| | | $\mathcal{L}_{RNP\_ada\alpha}$ | **0.31±0.01** | **0.13±0.01** | **-0.61±0.00** | **0.92±0.01** | **3.10±0.01** |
| ANP | context | $\mathcal{L}_{VI}$ | **1.38±0.00** | **1.38±0.00** | 0.65±0.04 | **1.38±0.00** | **4.14±0.00** |
| | | $\mathcal{L}_{RNP\_ada\alpha}$ | **1.38±0.00** | **1.38±0.00** | **0.97±0.11** | **1.38±0.00** | **4.14±0.00** |
| | target | $\mathcal{L}_{VI}$ | 0.81±0.00 | 0.64±0.00 | -0.91±0.02 | **1.06±0.01** | **3.65±0.01** |
| | | $\mathcal{L}_{RNP\_ada\alpha}$ | **0.83±0.01** | **0.66±0.01** | **-0.71±0.05** | **1.06±0.01** | **3.65±0.01** |
| TNPD | context | $\mathcal{L}_{VI}$ | **2.58±0.01** | **2.57±0.01** | **-0.52±0.00** | 1.73±0.11 | 10.63±0.12 |
| | | $\mathcal{L}_{RNP\_ada\alpha}$ | **2.58±0.01** | **2.57±0.01** | **-0.52±0.00** | **1.94±0.02** | **10.73±0.57** |
| | target | $\mathcal{L}_{VI}$ | 1.38±0.01 | **1.03±0.00** | **-0.59±0.00** | **1.63±0.07** | 6.69±0.04 |
| | | $\mathcal{L}_{RNP\_ada\alpha}$ | **1.39±0.00** | **1.03±0.00** | **-0.59±0.00** | 1.56±0.02 | **6.71±0.24** |

Gaussian as an approximate posterior $q(\theta) = \prod_i q(\theta_i)$ with $q(\theta_1) = \mathcal{N}(\mu_1, \lambda_1^{-1})$, , we can show the variance of the approximate posterior factorized Gaussian varies by alpha: $\lambda_1 = \rho_\alpha \Sigma_{11}$ where $\rho_\alpha = \frac{1}{2\alpha}\left[(2\alpha - 1) + \sqrt{1 - \frac{4\alpha(1-\alpha)\Sigma_{12}^2}{\Sigma_{11}\Sigma_{22}}}\right]$ is non-decreasing in $\alpha$.

### A.9. Prior misspecification settings

We consider two scenarios of prior misspecification: $q(\mathbf{z}|\mathbb{C}_{bad}, \varphi)$ and $q(\mathbf{z}|\mathbb{C}, \varphi_{bad})$. For $q(\mathbf{z}|\mathbb{C}_{bad}, \varphi)$, we design the experiments with the following setting: keep the target data $(X_T, Y_T)$ clean and corrupt the context data with noise $\tilde{y}_\mathbb{C} = (1 - \beta) * y_\mathbb{C} + \beta * \epsilon, \epsilon \sim \mathcal{N}(0, 1)$. The noise level $\beta$ is set as 0.3 for both training and testing, and the marginal predictive distribution is $p(Y_T|X_T, X_C, \tilde{Y}_C)$ and we report the test target set log-likelihood in Table 4.

For bad parameterization $q(\mathbf{z}|\mathbb{C}, \varphi_{bad})$. We adopted domain shift datasets since $p(\mathbf{z}|\mathbb{D}_{\text{train}})$ and $p(\mathbf{z}|\mathbb{D}_{\text{test}})$ do not come from the same distribution. Therefore, the prior model is suboptimal when conditioned on the training parameters $q(\mathbf{z}|\mathbb{C}, \varphi_{train})$.

### A.10. Tuning of $\alpha$.

We can start with training the model with the KL objective ($\alpha \to 1$) then gradually decrease (with granularity according to computational constraints) to 0. The intuition is inspired by KL annealing for VAE models Bowman et al. (2016), which starts with a strong prior penalization (close to 1) to reduce the posterior variance quickly and gradually reduces the prior penalization (close to 0) and focuses more on model expressiveness. The results are presented in Table 5. Our heuristics still managed to outperform the baselines across multiple datasets and methods.

### A.11. Additional Results.

We provided some qualitative results of the baseline NPs as well as their corresponding RNPs in Fig 5, 6, 7. Given the same context sets, we compare the results obtained by the original method on the left with the RNPs on the right. The number of context is chosen to be relatively small for RBF and Matern and more context points are selected for periodic to capture the periodicity of the function. RNPs tend to predict smaller variances for different function samples. While most methods struggle with the periodic dataset, ANP and VNP using our RNP objective significantly improved the baseline models.

Table 6 compares RNP with a simple baseline model using separate parameters of the prior and posterior models.

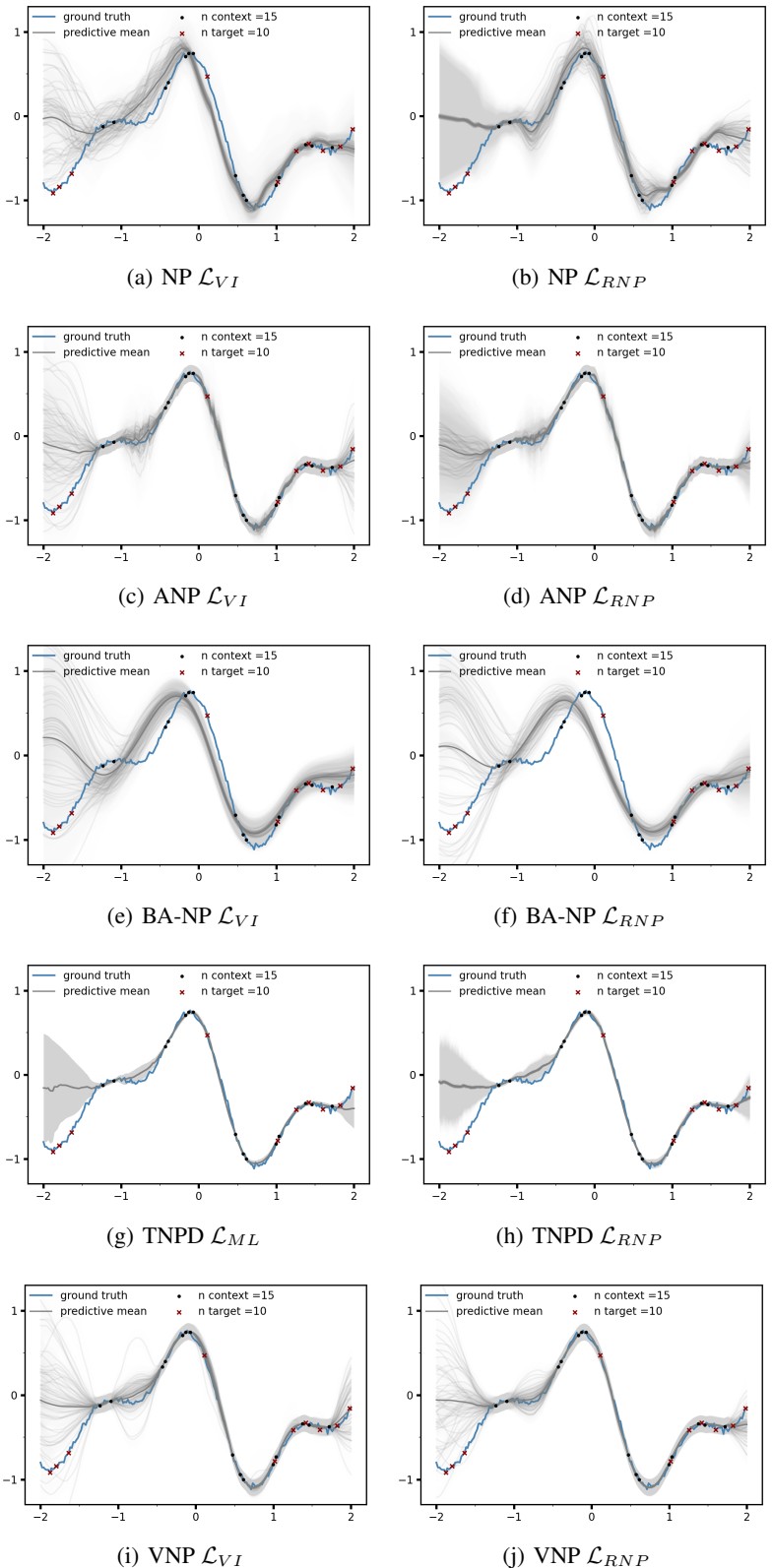

(a) NP $\mathcal{L}_{VI}$         (b) NP $\mathcal{L}_{RNP}$

(c) ANP $\mathcal{L}_{VI}$         (d) ANP $\mathcal{L}_{RNP}$

(e) BA-NP $\mathcal{L}_{VI}$         (f) BA-NP $\mathcal{L}_{RNP}$

(g) TNPD $\mathcal{L}_{ML}$         (h) TNPD $\mathcal{L}_{RNP}$

(i) VNP $\mathcal{L}_{VI}$         (j) VNP $\mathcal{L}_{RNP}$

*Figure 5.* 1D regression RBF dataset.

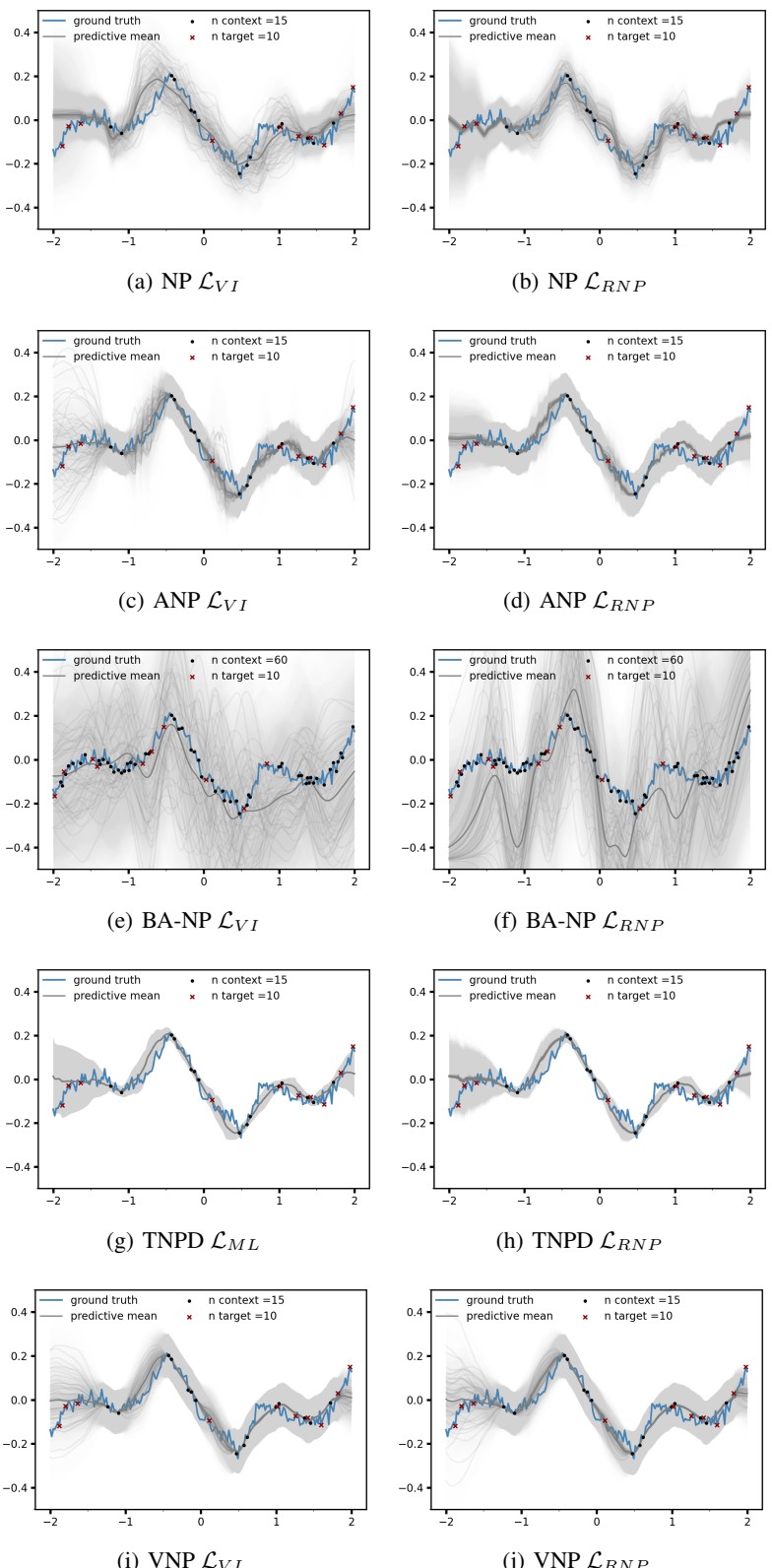

*Figure 6.* 1D regression Matern dataset.

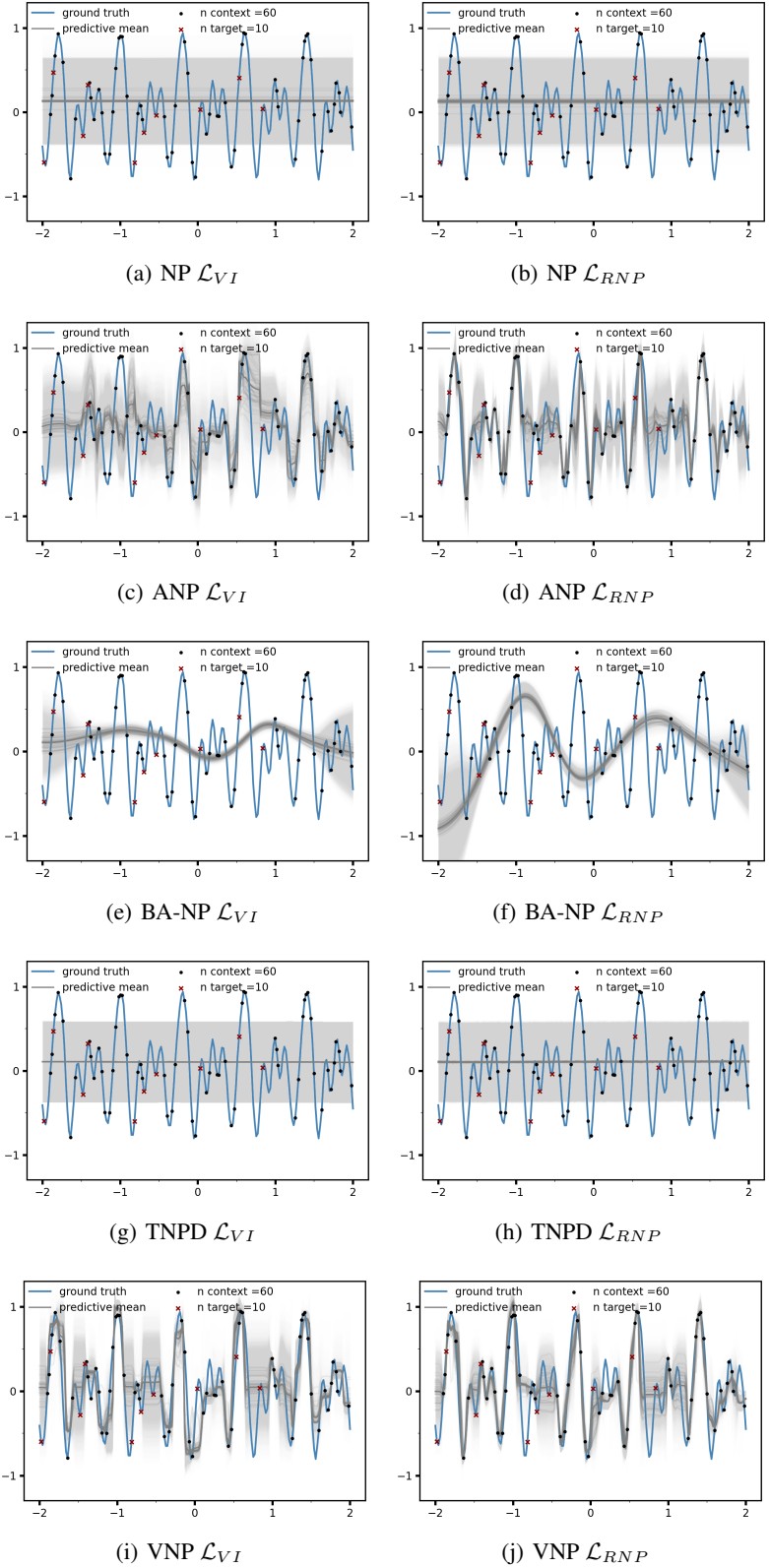

*Figure 7.* 1D regression Periodic dataset.

*Table 6.* Simple baseline comparison. In the setting column Separate PQ means the prior and posterior models are parameterised separately for the NP frameworks and optimised using the VI objective

| Model | Set | Setting | RBF | Matern 5/2 | Periodic | MNIST | SVHN |
|---|---|---|---|---|---|---|---|
| NP | context | Separate PQ | 0.56±0.02 | 0.41±0.01 | -0.50±0.00 | 1.00±0.03 | 3.21±0.01 |
| | | $\mathcal{L}_{RNP(\alpha)}$ | **0.78±0.01** | **0.66±0.01** | **-0.49±0.00** | **1.01±0.02** | **3.26±0.01** |
| | target | Separate PQ | 0.18±0.01 | 0.01±0.00 | **-0.61±0.00** | 0.90±0.02 | 3.05±0.01 |
| | | $\mathcal{L}_{RNP(\alpha)}$ | **0.33±0.01** | **0.16±0.01** | -0.62±0.00 | **0.91±0.01** | **3.09±0.01** |
| ANP | context | Separate PQ | **1.38±0.00** | **1.38±0.00** | -0.17±0.25 | **1.38±0.00** | **4.14±0.00** |
| | | $\mathcal{L}_{RNP(\alpha)}$ | **1.38±0.00** | **1.38±0.00** | **1.22±0.02** | **1.38±0.00** | **4.14±0.00** |
| | target | Separate PQ | 0.80±0.01 | 0.63±0.01 | -0.70±0.02 | **1.06±0.00** | **3.66±0.01** |
| | | $\mathcal{L}_{RNP(\alpha)}$ | **0.84±0.00** | **0.67±0.00** | **-0.57±0.01** | 1.05±0.01 | 3.61±0.02 |

*Table 7.* Wall clock time (in seconds) comparison. 32 MC samples were chosen during training to be consistent with the results in Table 1.

| Model | Setting | RBF | Matern 5/2 | Periodic | MNIST | SVHN |
|---|---|---|---|---|---|---|
| NP | $\mathcal{L}_{VI}$ | 1028 | 1015 | 1067 | 3231 | 4620 |
| | $\mathcal{L}_{RNP(\alpha)}$ | 1108 | 1000 | 1048 | 3058 | 4780 |
| ANP | $\mathcal{L}_{VI}$ | 1585 | 1620 | 1721 | 3770 | 5391 |
| | $\mathcal{L}_{RNP(\alpha)}$ | 1712 | 1693 | 1671 | 3605 | 5444 |

Table 7 added a wall clock time comparison between our objective and the VI objective.

