# OpenReview forum: "Rényi Neural Processes"
_ICML.cc/2025/Conference — ICML 2025 oral_

### Official Review · Reviewer_GkqA · 2025-03-09

**Overall Recommendation:** 4

**Summary:**

This paper identifies an important issue in Neural Processes (NPs) where prior misspecification can appear from the fact that the conditional prior and posterior share parameters, which can lead to  and degrade uncertainty estimates. The authors propose Rényi Neural Processes (RNPs) as a solution, replacing the KL divergence in the NP objective with the Rényi divergence. This modification introduces a tunable hyperparameter $\alpha$, which allows adjusting the influence of the prior and helps mitigate misspecification.

One of the key contributions of this work is showing how RNPs can bridge variational inference (VI) and maximum likelihood (ML) objectives. The paper also provides both theoretical support and empirical results. Experiments on datasets like MNIST, SVHN, and CelebA show improved log-likelihoods and more reliable predictions, particularly in cases where the prior is misspecified.

**Claims And Evidence:**

The main claims in the paper are:

1. Prior misspecification is an issue in NPs, and appears because of the parameter-sharing assumption.
2. Using Rényi divergence instead of KL divergence mitigates this issue by allowing more flexibility in how the prior affects the posterior.
3. The proposed approach unifies VI and ML objectives, offering a more general training framework for NPs.
4. RNPs consistently outperform standard NPs across a variety of tasks and settings.

These claims are well-supported both theoretically and experimentally. The paper presents detailed derivations and proofs, showing how the Rényi divergence modifies the posterior updates.

**Essential References Not Discussed:**

N/A

**Experimental Designs Or Analyses:**

The experiments are well designed and include a good mix of standard benchmarks. The main findings are:

1. RNPs achieve better log-likelihoods than baseline NPs.
2. RNPs improve performance in the presence of prior misspecification, particularly in cases with noisy or distribution-shifted contexts.
3. Tuning $\alpha$ appropriately leads to noticeable gains, but default values (e.g., $\alpha = 0.7$ for VI, $\alpha = 0.3$ for ML) seem to work well across tasks.

One strength of the experimental setup is the inclusion of ablation studies that analyze different aspects of RNPs, including Monte Carlo sampling and the number of context points.

**Methods And Evaluation Criteria:**

The authors test RNPs on:
1. 1D regression using different kernels.
2. Image inpainting tasks on MNIST, SVHN, and CelebA.
3. Ablation study with prior misspecification.

The experiments also include ablation studies to analyze the impact of Monte Carlo sampling, the number of context points, and the choice of $\alpha$. Overall, the methodology is well thought out and effectively demonstrates the strengths of RNPs.

**Other Comments Or Suggestions:**

Consider adding more intuition or visualizations to help explain why Rényi divergence improves posterior estimation. Also, a discussion on scalability would be useful, can this method be made more efficient for larger datasets?

**Other Strengths And Weaknesses:**

Strengths:
1. Identifies a misspedification problem in standard NPs and provides a simple but effective solution.
2. Theoretical contributions are rigorous and clearly explained.
3. Extensive experiments on multiple datasets and settings.
4. Ablation studies provide useful insights into the behavior of the method.

Weaknesses:
1. Computational cost. Monte Carlo sampling adds overhead, which could be an issue for large-scale applications.
2. Hyperparameter tuning. Tuning $\alpha$ is important, and the method might require careful tuning in some cases.

**Questions For Authors:**

1. Did you try other robust divergence, for example, $f$-divergence? If so, how did they compare?
2. Would RNPs work on larger datasets or real-time applications? Are there any known limitations?

**Relation To Broader Scientific Literature:**

This work builds on the Neural Processes (NP) family and connects it with ideas from robust divergence measures in variational inference. It references key prior work, including:

1. The original NP papers by Garnelo et al.
2. Extensions like Attentive NPs (ANP) and Transformer NPs (TNP)
3. Research on robust divergences like Rényi divergence and $\alpha$-divergence

The contribution is well-motivated, but the paper could benefit from a deeper discussion of alternative robust divergences (e.g., $\alpha$-divergence, $f$-divergence) and why Rényi divergence is particularly well suited for NPs.

**Theoretical Claims:**

The paper demonstrates that replacing KL divergence with Rényi divergence allows for more robust posterior updates while still maintaining a connection to standard NP objectives. One of the most interesting theoretical insights is that RNPs naturally interpolate between VI and ML objectives. By adjusting $\alpha$, the model can shift between behavior similar to KL-based variational inference ($\alpha \approx 1$) and maximum likelihood estimation ($\alpha \approx 0$).

---

> ### Author Rebuttal · Authors · 2025-03-31
>
> We thank the reviewer for acknowledging the effectiveness and theoretical rigor of our work. We appreciate their effort in helping us improve the efficiency and practicality of the work.
>
> # Computational costs.
>
> As shown in Supp Table 7 of the paper, we have already compared the wall clock time between our RNP and the VI objectives and no significant differences were observed. Our computational complexity is linear to the number of MC samples, which is also comparable to the VI objective. We will add more clarifications for efficiency in sec 5.4.
>
> # Hyperparameter tuning of $\alpha$ and adaptive $\alpha$ tuning.
>
> Since cross-validation can be computationally expensive, we have suggested the heuristics in sec 5.3 to gradually anneal $\alpha$ from 1 to 0 without hyperparameter tuning. We have already shown in Supp Table 5 that our RNP consistently outperformed competing approaches. We will elaborate further on this heuristic in sec 5.3.
>
> # Scalability on larger datasets or real-time applications.
>
> We would like to clarify that NPs are generally scalable to large datasets due to minibatch training and fast inference of the framework, and we have tested all our NP models on a sizeable image dataset CelebA. To further improve our efficiency, we suggest to adopt variance reduction methods such as control variates and Rao-Blackwellization [1] that could require fewer MC samples.  Additionally, for attention-based NPs, We suggest to use efficient attention mechanisms, e.g., Nyströmformer [2] which uses low rank approximation of the attention matrix. We will leave the efficiency improvement for our future work.
>
> [1] Ranganath, R. et al, 2014, Black box variational inference. In AISTAT. PMLR.
> [2] Xiong, Y. et al, 2021, Nyströmformer: A nyström-based algorithm for approximating self-attention. In AAAI.
>
> # $f$-divergence results.
> We added the comparing results using $f$-variational bound in [3]. More specifically, by specifying a convex function $f$ and its dual $f^*$, $f$-divergence connects several divergences including KL and Rényi divergences. Based on eq (8) in [3], we have the objective
>  $L_f(\phi, \theta) = \mathbb{E}_{q(z; \phi)}[f^*(\frac{p(z, Y_T|X_T, C; \theta)}{q(z; \phi)})] \geq f^* (p(Y_T|X_T, C))$.
>
> We chose the posterior $q(z;\phi)= q(z|C, T; \phi)$ like NPs do and compared our RNP objective with two functions.
>     $\chi^2$ divergence [3]:  $f(u)= \frac{1}{u} - u,   f^*(t)=t^2 -1$. and Jeffery divergence [4]:  $f(u) = (u-1)\log u,    f^*(t)=(t-1)\log t$.
>
> | Method |   Set   | Objective |      RBF      |   Matern 5/2  |  Periodic  |
> |:------:|:-------:|:---------:|:-------------:|:-------------:|:----------:|
> |   NP   | context | $\chi^2$  | 0.64±0.01     | 0.52±0.03     | -0.49±0.01 |
> |        |         | Jeffery   | -0.00±0.00    | -0.01±0.01    | -0.59±0.01 |
> |        |         | RNP       | **0.78±0.01** | **0.66±0.01** | -0.49±0.00 |
> |        |  target | $\chi^2$  | 0.21±0.02     | 0.07±0.01     | -0.68±0.00 |
> |        |         | Jeffery   | -0.23±0.00    | -0.27±0.00    | -0.61±0.00 |
> |        |         | RNP       | **0.33±0.01** | **0.16±0.01** | -0.62±0.00 |
>
> The results provide additional support that the RNP objective improves the baseline models. We also tested f-divergence minimization approaches based on the bound of Nguyen et al [5] such as that proposed in [4]. However, as with many GAN-type objectives, they led to unstable training. Finally, unlike our approach, it is unclear how f-divergence minimization methods proposed in the literature can be extended to  improve robustness of maximum likelihood-based NPs.
>
> [3] Wan, N. et al, 2020, F-divergence variational inference. NeurIPS.
>
> [4] Nowozin, S. et al, 2016, f-gan: Training generative neural samplers using variational divergence minimization. NeurIPS.
>
> [5] Nguyen, X. et al, 2010, Estimating divergence functionals and the likelihood ratio by convex risk minimization. IEEE Transactions on Information Theory.
>
> # Intuition or visualizations for RNP posterior estimation.
>
> In Fig 1 (a) we have visualized the two posterior distributions obtained by our RNP and VI objectives. The VI objective produced an overestimate of the posterior variance, indicating a strong regularization from the prior model. As a result, it fitted the periodic data with a large variance and over-smoothed mean functions as shown in Fig 1 (b), whereas RNP dampens the prior regularization and obtained a posterior with much smaller variance and therefore encourages the likelihood model to be more expressive as illustrated in Fig 1 (c).

---

### Official Review · Reviewer_3i1T · 2025-03-13

**Overall Recommendation:** 4

**Summary:**

The paper presents a novel approach for training neural processes. By replacing the conventional KL divergence with the Renyi divergence, this allows the model to adapt when confronted with a misspecified prior, therefore enabling more robust inference. This paradigm is somewhat analogous to the utilisation of a hierarchical prior. Experiments are conducted on a mix of regression and image tasks, showing improvement in log likelihood performance over existing techniques.

**Claims And Evidence:**

The claims made are generally well supported by the empirical evidence

**Essential References Not Discussed:**

When introducing the Renyi divergence on page 2, it would be appropriate to cite the paper which first proposed it, which was Renyi 1961.

**Experimental Designs Or Analyses:**

The experiments span a good range of datasets and in most cases are presented alongside suitable uncertainty estimates.

In Tables 1 and 2, currently only the numerically highest performing log likelihood is set in bold, but in many cases there is no statistically significant difference to the second strongest method. I would recommend applying bold only to results that outperform at a statistically significant level (and explicitly state that chosen level). This is particularly significant in Table 2 where the L_ML is bolded twice when in neither case it is significantly superior.

The captions are a little sparse, for example Figure 3 simply reads "Hyperparameter tuning" and Figure 4 is "Ablation study". I'd recommend ensuring that the captions to the tables and figures are self sufficient.

Figure 3 is somewhat lacking in uncertainty quantification, presumably this is showing the outcome for only a single split. It's therefore not clear how much of the functional form is stochastic in nature.

**Methods And Evaluation Criteria:**

Yes, a suitable selection of benchmarks and NP architectures are shown.

**Other Comments Or Suggestions:**

Figure 3 seems to illustrate alpha going up to 2.0 but in the text it is mentioned that it is restricted to alpha<1. Perhaps this figure will be more informative if we can highlight (as a horizontal dotted line for example) the vanilla alpha=1 value. I would also recommend maintaining a similar dynamic range on the y axis, between different panels, as this will illustrate eg that the vanilla NP on RBF is much more sensitive to the choice fo alpha.

It might be of interest to comment on the relation between using Renyi divergence and using a hierarchical prior. In that the latter one would explicitly evaluate a range of priors, while in the Renyi case the alternative forms of the prior are implicit.

And a small typo:
"The limitations of our framework lie in drawing multiple samples...." presumably ought to read
"A limitation of our framework lies in drawing multiple samples...."

**Other Strengths And Weaknesses:**

A novel, well motivated and clearly presented paper

I think the main weakness relates to how some of the experimental results are missing details. It's not clear to me why Figure 2 shows just the Lynx data but not the Hare. Do the results in Table 2 relate to the data from both species or is it also just for the Lynx?

**Questions For Authors:**

No further questions at this stage!

**Relation To Broader Scientific Literature:**

This work seeks to build on previous studies in the development of Neural Processes.

**Theoretical Claims:**

No, while some theoretical background is given, the key results here are empirical.

---

> ### Author Rebuttal · Authors · 2025-03-31
>
> We appreciate the reviewer's efforts in helping us refine the details and acknowledge the original literature. We agree that a rigorous analysis and self-sufficient figures would strengthen the soundness of our work.
>
> # Significance tests in Table 1 and 2.
>
> All the results presented in the paper were reported with error bars (i.e., standard deviations). As the reviewer suggested, we performed two‐sample t-tests between our RNP and the second best method and highlighted the bold results with significant improvements of **p value $<$ 0.05**. We observed that in Table 1 we significantly improved RBF, Matern and Periodic datasets for most of the methods. Here we show the updated Table 2. In Table 2, the ML objective is no longer significantly superior than our RNP objective on the  D\_train EMNIST dataset.
>
> | Objective | D_train  (Lotka-Volterra) |               | Misspec D_test  (Hare-Lynx) |                | D_train EMNIST  (class 0-10) |           | Misspec D_test  (class 11-46) |               |
> |:---------:|:-------------------------:|:-------------:|:---------------------------:|:--------------:|:----------------------------:|:---------:|:-----------------------------:|:-------------:|
> |           |          context          |     target    |           context           |     target     |            context           |   target  |            context            |     target    |
> |    L_ML   |         3.09±0.22         |   1.98±0.11   |          -0.59±0.47         |   -4.44±0.41   |           1.54±0.05          | 1.56±0.07 |           0.03±0.97           |   -0.20±0.57  |
> |   L_RNP   |         3.32±0.15         | **2.12±0.06** |          -0.17±0.31         | **-3.63±0.09** |           1.52±0.08          | 1.47±0.12 |           0.96±0.18           | **0.70±0.15** |
>
> # Missing details for the Hare Lynx dataset.
>
> We apologize for the missing details. Table 2 in the paper reported the results for both species as we treat them as a 2-dimensional input system with feature correlations (see sec 5.2). We left out the Hare plots in Fig 2 originally because when we tried to plot both the Lynx and Hare results in a single graph, the uncertainty intervals of two species were heavily overlapped and impeded the visibility. We will add the Hare results using different colors and opacity in the final version. We will also add more descriptions of this dataset in sec 5.2 and supplementary section for better clarification.
>
> # Relationship to hierarchical prior models.
>
> We thank the reviewer for raising this interesting point. We believe  our approach to misspecification in neural processes based on the Rényi divergence is fundamentally different to hierarchical approaches. By introducing additional latent variables in a hierarchical fashion, one aims to have a more flexible marginal prior model. For example, under some mild conditions, mixture models are known to be able to approximate any continuous density. However, usually, that additional flexibility comes at the cost of more complex inference. Our approach, maintains the original prior but deals with model misspecification through dampening with the $\alpha$ parameter. As pointed out by Reviewer Fmj6, it is more closely related to robust Bayesian inference methods based on Gibbs/tempered posteriors. We will discuss this in the final version.
>
> # Updating figures, captions, typos and references.
>
> - **Captions of Fig 3 and Fig 4.** We will add captions for them to be self-sufficient. Fig 3:  cross-validation is used to select the optimal $\alpha \in [0, 2]$. Fig 4: We investigated how MC sample sizes and the number of context points affect test log-likelihood.
>
> - **Uncertainty quantification in Fig 3.** We have actually plotted the uncertainty intervals but some of them were occluded by thicker lines of mean values (see Fig 3 (a) TNPD $\alpha = 1.5$ for better visibility). We will change the color and the thickness of the intervals to make them more evident.
>
> - **$\alpha$ range and dynamic plot range in Fig 3**. Fig 3 showed that $\alpha >1$ impedes NP training due to an overestimate of the posterior variance (please check the comment for Reviewer aNVZ ``The effect of $\alpha$ values on training'' for more details). Therefore, we recommend tuning $\alpha \in (0, 1)$ in the text. As suggested by the reviewer, we will highlight the results using the KL objective which corresponds to $\alpha = 1$ in the plot and change the plot into a dynamic range.
>
> - **Adding reference**. We will add the original Rényi divergence paper for reference.
>
> - **Correcting typos**. We will fix the typo in the limitation discussion and proof read the paper.

---

> > ### Comment · Reviewer_3i1T · 2025-04-08
> >
> > Thank you for the detailed response, I'm glad the feedback was helpful, and that Tables 1 & 2 have been strengthened.
> >
> > With regards to the hierarchical prior - my suspicion here was that for any given value of alpha (and for a given dataset), there exists an implicit alternative prior that would generate the same posterior as that value of alpha. Thus marginalising or tuning alpha could be deemed equivalent to marginalising or optimising a hyperprior. (I only mention this as it might counteract any criticism or concerns that this approach - as with tempered posteriors - is no longer fully Bayesian.)

---

> > > ### Author Response · Authors · 2025-04-08
> > >
> > > Dear Reviewer,
> > >
> > > Thank you for increasing your initial score. We sincerely appreciate your helpful feedback and thoughtful defense of our work. We will make sure to discuss the hyperprior on alpha in the revision.
> > >
> > > Sincerely,
> > >
> > > The Authors

---

### Official Review · Reviewer_Fmj6 · 2025-03-13

**Overall Recommendation:** 4

**Summary:**

The paper introduces Rényi Neural Processes (RNPs), a modification of Neural Processes (NPs) that replaces the standard Kullback-Leibler (KL) divergence with the Rényi divergence to mitigate prior misspecification. The authors argue that parameter coupling between the prior and posterior in traditional NPs leads to biased variance estimates and propose RNPs as a more flexible alternative. The method is tested on regression and image inpainting tasks, showing improved log-likelihood performance compared to state-of-the-art NP variants.


_(for any missing input on any of the fields, please refer to the **Strengths and Weaknesses** or the **Other comments or suggestions** sections)_

**Claims And Evidence:**

- The parameter coupling in standard NPs leads to prior misspecification and degraded performance.
  - *Evidence*: Theoretical derivations and empirical evaluations demonstrate that standard NPs overestimate posterior variance, leading to oversmoothed predictions.
- Using Rényi divergence provides a tunable mechanism to reduce the impact of prior misspecification.
  - *Evidence*: The proposed RNPs consistently outperform standard NPs and other NP variants across multiple benchmarks.
- RNPs improve generalization without modifying model architectures.
  - *Evidence*: The method is applied to existing NP models (ANP, VNP, TNP-D) with measurable improvements.

**Essential References Not Discussed:**

- A deeper discussion on the connection to PAC-Bayes approaches and other Bayesian robustness techniques could strengthen the theoretical grounding.
- Explicit comparisons with f-divergence-based variational inference methods would be useful.
- Regarding the choice of $\alpha$, the authors could consider mentioning [1], which discusses the impact of $\alpha$ in variational inference.
- Related to the previous work, [2] seems strongly related to this contribution since they show that more flexible models in combination with robust divergences may fix prior misspecification issues. Not essential, although maybe worth mentioning here.
- Some references to robust inference outside the NP community could provide a broader perspective. As a suggestion, the authors may consider mentioning other works that make use of robust divergences in Bayesian inference, like [3] (which, for instance, also mentions NPs as a particular case of their model).

[1] Rodríguez-Santana, et al. "Adversarial $\alpha$-divergence minimization for Bayesian approximate inference." Neurocomputing 471 (2022): 260-274.

[2] Santana et al. "Correcting Model Bias with Sparse Implicit Processes." ICML 2022 Workshop "Beyond Bayes: Paths Towards Universal Reasoning Systems" arXiv preprint arXiv:2207.10673 (2022).

[3] Ma, et al. (2019, May). "Variational implicit processes". In International Conference on Machine Learning (pp. 4222-4233). PMLR.

**Ethical Review Concerns:**

The paper mentions potential misuse in reconstructing missing images, which could raise privacy concerns. Other than that, no immediate ethical risks beyond standard considerations in probabilistic modeling. In fact, more accurate uncertainty estimates could lead to more responsible decision-making in AI applications.

**Experimental Designs Or Analyses:**

- Well-structured with comprehensive baselines.
- Includes ablation studies on hyperparameter tuning ($\alpha$ selection) and Monte Carlo sample size.
- Tests both parameterization-induced and context-induced prior misspecification.
- Considers real-world applications (Hare-Lynx dataset for time series forecasting).

**Methods And Evaluation Criteria:**

- The primary modification is substituting KL divergence with Rényi divergence in the NP objective.
- Evaluations focus on predictive log-likelihood, testing the approach on 1D Gaussian process regression and image inpainting (MNIST, SVHN, CelebA).
- Comparison against NP, ANP, VNP, and other baselines.
- Additional experiments assess robustness under prior misspecification (e.g., noisy context points, domain shifts).

**Other Comments Or Suggestions:**

- Investigating adaptive $\alpha$ selection methods could reduce the need for cross-validation.
- Notation can sometimes be cumbersome (e.g. Eqs 3, 5 and 6). Maybe it would be beneficial to simplify it for readability.


EDIT: I increased my initial score after reading the authors' response.

**Other Strengths And Weaknesses:**

**Strengths:**
- Well-motivated theoretical derivations to bolster the theoretical claims made in the text. The supplementary work provides a great deal of detail on the needed calculations needed to understand the method.
- The approach is clearly motivated and well-explained, with a strong connection to prior work.
- The implementation of the method is straightforward and could be integrated into existing NP frameworks.
- The proposed approach provides strong empirical results across multiple benchmarks.
- The paper is well-structured and written, making it easy to follow.

**Weaknesses:**
- Although the empirical results are strong, I fear that the initial idea of the paper might be a bit incremental. It is aided by the extensive theoretical derivations provided, but nonetheless, the core idea of using Rényi divergence to fix prior misspecification is not really that novel (similar efforts with similar empirical results have been achieved, e.g. see [1]).
- There is no clear way to choose the optimal value of  $\alpha$ for a given problem since it is dataset-dependent. This could be a limitation in practice, requiring cross-validation.

**Questions For Authors:**

1. In general, what are the implications inside Bayesian modelling of constructing the prior in a data-dependent fashion? Does this not imply a form of overfitting? Maybe a discussion on this and its relationship with extensions of the Bayesian inference framework as in the (already mentioned in the paper) Generalized Variational Inference.
2. Following the previous question, what theoretical guarantees can be provided on the inference results for the proposed method? In particular, I wonder about the soundness of the inference process and the quality of the uncertainty estimates.
3. How does the choice of $\alpha$ affect uncertainty calibration in practice?

**Relation To Broader Scientific Literature:**

- Builds on extensive prior work in Neural Processes, Variational Inference, and robust divergences. It seems to cover the most relevant literature for the proposed method.
- Connects with literature on robust Bayesian inference and alternative divergences (e.g., $\alpha$-divergence, f-divergence).

**Theoretical Claims:**

- "New insight into prior misspecification in NPs through the lens of robust divergences".
- Proves that the standard NP prior model is misspecified due to parameter coupling.
- Establishes that Rényi divergence generalizes KL divergence and allows tuning of prior penalization.
- Demonstrates that RNPs unify variational inference (VI) and maximum likelihood (ML) approaches, bridging two common objectives.

---

> ### Author Rebuttal · Authors · 2025-03-31
>
> We thank the reviewer for acknowledging our "well-motivated theoretical derivations" and "strong empirical results".
> # Incremental novelty
> We would like to clarify to the reviewer that our work goes beyond using RD for prior misspecification. We are the first to identify prior misspecification in the realm of NPs due to the parameter coupling between the prior conditional and the approximate posterior, and our theoretical and empirical analysis justify the use of robust divergences. Secondly, we introduce a new objective that unifies the VI and MLE objectives for NPs.
> # Adaptive $\alpha$ value selection
> Please kindly refer to the comments for the Reviewer GkqA for this and f-divergence related questions.
> # Results on uncertainty calibration
> We added the results of continuous ranked probability scores (CRPS) w.r.t different α values for NP and ANP on the RBF dataset. CRPS is a commonly used to measure the forecast accuracy and lower scores indicate better calibration. The results showed that the CRPS calibration improved when α increases from 0 to 0.9.
> | α | NP            | ANP           |
> |----------|---------------|---------------|
> | 0.0      | 0.1427±0.0016 | 0.0978±3e-4 |
> | 0.3      | 0.1452±0.0029 | 0.0942±2e-4 |
> | 0.7      | 0.1370±0.0018 | 0.0827±5e-4 |
> | 0.9      | 0.1321±0.0014 | 0.0815±5e-4 |
> | 1.0      | 0.1484±0.0023 | 0.0965±3e-4 |
> # Theoretical discussions
> We sincerely appreciate the reviewer's thoughtful feedback. While some of the questions raised by the reviewer extend beyond the immediate scope of this paper and address broader aspects of NP research, we recognize their importance and agree these points warrant further investigation. We are delighted to make every effort to clarify their concerns.
> - **Robust Bayesian inference and PAC-Bayes**
>
> Indeed, the literature on robust Bayesian inference is rich and we thank the reviewer for pointing out the need to discuss this. In particular, the relation to PAC-Bayes is fascinating. In short, dealing with model misspecification from a Bayesian perspective appears in the literature under several disguises, for example,  Gibbs posteriors, tempered posteriors and fractional posteriors [1,2]. These approaches essentially weigh the likelihood to temper its influence. Connections with PAC-Bayesian bounds controlling generalization in statistical learning have also been explored previously, more notably in [2]. These connections are extended  by explicitly analyzing variational approximations of PAC-Bayes/Gibbs posteriors [3].
> We will expand on this in the final version. However, it is important to emphasize that our focus is on neural processes but these approaches remain exciting directions for future work.
>
> [1] Grunwald, P, et al. Inconsistency of Bayesian inference for misspecified linear models, and a proposal for repairing it. Bayesian Analysis, 2017.
>
> [2] Bhattacharya, A., et al. Bayesian fractional posteriors. The Annals of Statistics, 2019.
>
> [3] Alquier, P., et al. On the properties of variational approximations of Gibbs posteriors. JMLR, 2016.
> - **Theoretical guarantees of RNP**
>
> The theoretical soundness of our work can be built on two parts: the frequentist consistency properties of the vanilla NPs using the KL divergence, and the relaxation of assumptions with our Rényi divergence. For the first part, we kindly refer the reviewer to section 3.3 -3.5 of the thesis [4] for prediction map approximation theory of NPs. Briefly, in the limit of infinite data, a variational family of prediction maps can recover the mean map of the true NP and the sum of the variance map and observation noise under some regularity assumptions. In the case of limited data, more assumptions including the input size, the compactness of the variational family, the boundedness of the data, and the boundedness of the stochastic process are required to guarantee the consistency. For the second part, some milder regularity conditions than KL divergence can be made for consistency [5], including a uniformly bounded prior distribution and a locally asymptotically normal likelihood model.
>
> [4] Bruinsma, W. (2022) Convolutional Conditional Neural Processes. Apollo - University of Cambridge Repository.
>
> [5] Jaiswal, P., Rao, et al, 2020. Asymptotic Consistency of α-Rényi-Approximate Posteriors. JMLR, 21(156).
> - **Data-dependent priors**
>
> We agree with the reviewer that it might sound un-Bayesian to have data-dependent priors, but in the case of NPs it is not only valid but necessary when viewed through the lens of hierarchical Bayes or meta-learning: the prior is being drawn from a hyper-prior. When we condition on context data, we're effectively doing Bayesian inference over a latent variable that defines the function.
> # Fixing typos and adding references
> We will add the recommended references as well as [6] to the main text. We will also simplify some equation notations.
>
> [6] Rodríguez-Santana et al, 2022. Function-space Inference with Sparse Implicit Processes. ICML.

---

### Official Review · Reviewer_aNVZ · 2025-03-15

**Overall Recommendation:** 4

**Summary:**

The paper replaces the Kullback–Leibler (KL) divergence in the standard neural processes (NPs) with the Renyi divergence to mitigate the issue of prior misspecification. The proposed Renyi neural process (RNP) has a tuning parameter $\alpha>0$ that penalizes the misspecified prior and unifies the variational inference ($\alpha=1$) and maximum likelihood estimation ($\alpha=0$) in the same framework. The reason of prior misspecification in NP is explained and the mitigation by RNP is well illustrated. The paper further investigates the robustness of Renyi divergence applied to other state-of-the-art (SOTA) NP algorithms and demonstrates advantages of Renyi divergence in NP applications.

**Claims And Evidence:**

The claims are well explained and there is enough numerical evidence to support the claims.

**Essential References Not Discussed:**

None.

**Experimental Designs Or Analyses:**

Most of the experiments are well designed. I do have following questions:


1. Figure 1: what controls the smoothness and the correlation strength of RNP, all by $\alpha$? BTW, the labels in caption are wrong: '(c)' should be '(b)' and '(d)' should be '(c)'.
2. Figure 2: What is the advantage of RNP (b) over VI (a) in ANP? The estimates with similar smoothness and both miss the truth between 0.5 and 1.
3. How does $\alpha$ affect the training? Is there any particular value that poses challenges in training, e.g. large gradients, slow convergence?

**Methods And Evaluation Criteria:**

The paper includes comprehensive numerical studies. However, they are focused only on the objective function. It would be nice to include other metrics, e.g. relative error comparing against the truth, in one of the comparisons.

**Other Comments Or Suggestions:**

Two sentences on page 5 under "Robust divergence" seem incomplete:

"KL divergence ... between the posterior..." between the posteriors or between the posterior and the prior?
"Several other ... features are noise or the existence of outlier."?

**Other Strengths And Weaknesses:**

The application of Renyi divergence to neural process is novel. The contribution that unifies VI and ML is appreciated.

**Questions For Authors:**

See above.

**Relation To Broader Scientific Literature:**

The paper provides  a good contribution to neural process algorithms by exploring a divergence more robust to prior misspecification.

**Theoretical Claims:**

Yes. The proofs appear correct.

---

> ### Author Rebuttal · Authors · 2025-03-31
>
> We thank the reviewer for acknowledging our innovation of unifying the objectives and carefully reviewing details including supplementary materials and numerical results.
>
> # Additional evaluation metrics.
>
> We have additionally reported the relative errors for two baseline methods NP and ANP on three GP regression datasets. Our objective still generally outperformed baseline models on this new metric. We will incorporate more results in the  version.
>
> | Method |   Set   | Objective |      RBF      |   Matern 5/2  |    Periodic   |
> |:------:|:-------:|:---------:|:-------------:|:-------------:|:-------------:|
> |   NP   | context | L\_VI     | 2.56±0.12     | 5.79±0.73     | 2.88±0.09     |
> |        |         | L\_ML     | 2.47±0.14     | 4.63±1.42     | 2.87±0.08     |
> |        |         | L\_RNP    | **2.09±0.10** | **4.48±1.16** | **2.77±0.10** |
> |        |  target | L\_VI     | 4.80±0.40     | 3.59±0.12     | 6.42±0.35     |
> |        |         | L\_ML     | 4.52±0.36     | 3.33±0.07     | 6.42±0.43     |
> |        |         | L\_RNP    | **4.23±0.35** | 3.40±0.16     | **6.22±0.25** |
> |   ANP  | context | L\_VI     | 0.17±0.01     | 0.28±0.03     | 2.76±0.39     |
> |        |         | L\_ML     | 0.18±0.01     | 0.24±0.03     | 2.92±0.38     |
> |        |         | L\_RNP    | 0.17±0.01     | 0.25±0.06     | **0.52±0.08** |
> |        |  target | L\_VI     | 2.51±0.06     | 2.42±0.05     | 8.81±0.43     |
> |        |         | L\_ML     | 2.55±0.07     | 2.40±0.04     | 8.27±0.86     |
> |        |         | L\_RNP    | **2.09±0.05** | **1.89±0.03** | **6.15±0.31** |
>
>
> # The effect of $\alpha$ values on training.
>
> $\alpha > 1$ usually impedes training as it focuses too much on improving the mass-covering of the posterior, resulting in an overestimate of the posterior variance (Please refer to Fig 3 in the paper for details).  Regarding the convergence rate, [1] showed that under mild regularity assumptions, it is bounded by $\sqrt{n}$ with n being the number of samples rather than by $\alpha$ values.
>
> [1] Jaiswal, P., Rao, et al, 2020. Asymptotic Consistency of $\alpha $-Renyi-Approximate Posteriors. JMLR, 21(156).
>
>
> # Missing clarifications and typos.
>
> -  **Smoothness in Fig 1**.
> The difference between L_RNP and L_NP on smoothness and correlation strength is only controlled by $\alpha$. Due to the misspecified prior regularization in standard KL, vanilla ANPs (Fig 1b) struggle with over-smoothed predictions. Our objective dampens such regularization and encourages a more expressive likelihood model that better fits the data.
>
> - **RNP advantage in Fig 2**.
>     RNP indeed did not impose a significant advantage over VI as shown in Fig 2 (a) and (b). However, our main claim in Fig 2 is that RNP achieves better uncertainty estimate than the ML objective.
>
> - **Typos and missing captions**.
> We thank the reviewer for pointing out the typos in the captions of Fig 1, which we will correct in the revision.  We will also add explanations for Fig 5, 6, 7 in the supplementary and fix the typos in sec 4 related work.

---

### Decision · Program_Chairs · 2025-05-01

**Decision:**

Accept (oral)

**Comment:**

All reviewers agreed that this paper addresses an important topic and is well motivated. There is a consensus that the claims are both valuable and well supported by extensive experimentation.
The authors addressed all the questions and issues raised by the reviewers in the rebuttal.

From the reviews it seems that the paper can make an important contribution to the community by:
1. Identifying gaps in current practice of NPs and provide a proposed solution.
2. The theory of the proposed method is well explained.
3. The method can be used in practice by integrating it into existing NP frameworks.

Some reviewers raised the concern that the method itself (using Rényi divergence to fix prior misspecification) is not novel but rather its integration within the NP framework.